# Radiomic tractometry reveals tract-specific imaging biomarkers in white matter

Peter Neher [1,2,3] ✉, Dusan Hirjak [4] & Klaus Maier-Hein[1,2,3,5]

Tract-specific microstructural analysis of the brain's white matter (WM) using diffusion MRI has been a driver for neuroscientific discovery with a wide range of applications. Tractometry enables localized tissue analysis along tracts but relies on bare summary statistics and reduces complex image information along a tract to few scalar values, and so may miss valuable information. This hampers the applicability of tractometry for predictive modelling. Radiomics is a promising method based on the analysis of numerous quantitative image features beyond what can be visually perceived, but has not yet been used for tract-specific analysis of white matter. Here we introduce radiomic tractometry (RadTract) and show that introducing rich radiomics-based feature sets into the world of tractometry enables improved predictive modelling while retaining the localization capability of tractometry. We demonstrate its value in a series of clinical populations, showcasing its performance in diagnosing disease subgroups in different datasets, as well as estimation of demographic and clinical parameters. We propose that RadTract could spark the establishment of a new generation of tract-specific imaging biomarkers with benefits for a range of applications from basic neuroscience to medical research.

A key element in understanding healthy and impaired brain structure and function is the analysis of its neural pathways, the white matter (WM). Over the last decades, the development of diffusion-weighted magnetic resonance imaging (dMRI) has revolutionized our ability to study WM in vivo. By probing the movement of water molecules, dMRI provides information about the microstructure, integrity, and connectivity of WM tracts. Many highly influential studies based on dMRI have been published, analyzing the WM and its alterations to gain insights into brain development, aging, injuries, and diseases or to study normal brain structure and function[1–8].

Analysis of brain-MRI data has progressed from global histogram-based analysis[9,10], over voxel-based statistical analysis matching individual subjects using registration algorithms[11,12], to WM-skeleton-based analysis of whole-brain data using tract-based spatial statistics (TBSS)[13]. While used frequently, these techniques have several limitations

discussed extensively in the literature[13–15]. Recently, there has been a shift towards tract-specific approaches based on fiber tractography[16–18]. These methods enable the targeted investigation of WM microstructure in the form of dMRI-derived parameter maps, such as the fractional anisotropy (FA) or mean diffusivity (MD)[19], within specific tracts.

Tract-specific analysis itself has evolved from studying tract averages[17,18,20] to an analysis of microstructural parameters along individual tracts[21–24]. Tract averages involve the calculation of statistics over the entire tract, which can be useful for investigating global changes in tract integrity. However, it does not provide information about regional variations within the tract, which are quite significant. In contrast, along-tract analysis (tractometry) involves dividing the tract into smaller parcels along its course and analyzing the diffusion metrics in each parcel separately. It is based on models of individual

[1]German Cancer Research Center (DKFZ) Heidelberg, Division of Medical Image Computing, Im Neuenheimer Feld 223, 69120 Heidelberg, Germany. [2]German Cancer Consortium (DKTK), partner site Heidelberg, Heidelberg, Germany. [3]Pattern Analysis and Learning Group, Department of Radiation Oncology, Heidelberg University Hospital, Heidelberg, Germany. [4]Department of Psychiatry and Psychotherapy, Central Institute of Mental Health, Medical Faculty Mannheim, Heidelberg University, J5, 68159 Mannheim, Germany. [5]National Center for Tumor Diseases (NCT), NCT Heidelberg, a partnership between DKFZ and the university medical center Heidelberg, Heidelberg, Germany. ✉e-mail: p.neher@dkfz-heidelberg.de

WM tracts obtained using fiber tractography[16]. The tract models consist of individual fibers, or streamlines, each of which is a series of 3D points that define its trajectory. To analyze the image along the tract, it is evaluated at these points and each value is assigned to one of $n$ parcels depending on its position. The values within each parcel are then aggregated, usually by averaging, resulting in a vector of scalar values along the tract that can be used for further analysis. This method provides a detailed picture of variations within the tract, allowing for the investigation of localized tissue alterations and specific functions associated with different parts of the tract. Tractometry has been used extensively in a variety of applications and can be considered as the state-of-the-art in tract-specific WM analysis[24–33].

Nevertheless, tractometry as well as other techniques widely used in neuroscience such as voxel-based analysis or TBSS are designed for group-level statistical analysis and only allow limited or no statements at the individual subject-level. In case of tractometry, we hypothesize that one reason for this is the drastic reduction of the complex image information along the tract to only few scalar values. This can result in the loss of information about variations and patterns along the tracts that might be crucial for subject-level predictions.

In other radiological domains, the concept of radiomics is a widely used approach for imaging-based tissue analysis[34,35]. The fundamental idea behind radiomics is that medical images contain a wealth of information beyond what is evident to the naked eye or what can be captured with simple scalar measures such as mean signal intensities or structure diameters. It involves the extraction and analysis of a large number of quantitative features from medical images that quantify subtle variations in pixel intensities, textures, shapes, and spatial relationships within an image. Radiomics has shown to yield valuable insights and promising results for subject-level predictions in various tasks, such as automated diagnosis, patient stratification, risk assessment and response monitoring[36,37]. In the context of brain imaging, radiomics has been used extensively for the analysis of tumors[38,39] and also for studying psychiatric and neurodegenerative diseases[40–43]. Nevertheless, the concept of radiomics has not yet found its way into the domain of tract-specific WM analysis.

Radiomics and tractometry are orthogonal approaches, in the sense that tractometry is focused on the localization of changes using a reduced feature set, while radiomics is focused on extracting as much information from an image region as possible and on providing advanced biomarkers, e.g. for predictive machine learning (ML), without specific focus on the localization of changes. We show that combining both approaches, by introducing rich radiomics-based feature-sets into the world of tractometry enables improved predictive modeling on the basis of individual WM tracts and also provides localization of tract regions that are most informative for the respective task. We call this approach radiomic tractometry (RadTract) and demonstrate its capabilities on four distinct datasets comprising individuals with Alzheimer's disease, Mild Cognitive Impairment, Parkinson's disease, prodromal Parkinson's disease, schizophrenia, and catatonia as well as matched healthy controls. RadTract markedly outperforms classic tractometry in diagnosing disease subgroups in all datasets. Using RadTract further yields promising results in estimating demographic and clinical parameters, such as age, education, symptom severity, or daily medication dose.

Overall, our results indicate that even well-studied parameter maps, such as the FA, contain a wealth of information that could be valuable for a wide range of applications, but that is currently lost for the state-of-the-art.

We anticipate this work to be a starting point for the development of a new generation of tract-specific imaging biomarkers, enabling not only better neuroscientific studies by providing a much more detailed view on microstructural patterns of WM tracts and their changes, but serving as a first step towards improved predictions on a subject-level. RadTract is also not limited to dMRI and WM analysis, but it is generally applicable to all kinds of imaging contrasts and also to all kinds of research questions involving fibrous tissue and tractography thereof, such as the analysis of microstructural properties of muscular tissue or the neurovascular anatomy of the prostate[44,45].

RadTract is available on GitHub (https://github.com/MIC-DKFZ/radtract) and as a ready to use python package (https://pypi.org/project/radtract/).

## Results

We present results obtained on four datasets, including data obtained from the Alzheimer's Disease Neuroimaging Initiative (ADNI, www.adni-info.org/), the Parkinson's Progression Markers Initiative (PPMI, www.ppmi-info.org/access-data-specimens/download-data, RRID: SCR_006431), the UCLA Consortium for Neuropsychiatric Phenomics LA5c Study (SCHZ, https://openfmri.org/dataset/ds000030/)[46] and a non-public dataset (CAT) acquired at the Central Institute of Mental Health (CIMH, https://www.zi-mannheim.de/en/)[27]. These distinct datasets enabled a broad range of experiments on imaging data of healthy and diseased individuals, well suited to demonstrate the capabilities and general applicability of our approach. A total of 46 WM tracts were investigated individually. Features were calculated from four widely used tensor-based parameter maps, namely FA, apparent diffusion coefficient (ADC), axial diffusivity (AD) and radial diffusivity (RD). As baseline methods, we used two variations of classic tractometry, Centerline Tractometry and Static Tractometry, which are state-of-the-art for tract-specific analysis. Other widely used techniques, such as voxel-based analysis or TBSS were not included as benchmarks, since they are designed for purely global group-level analysis and do not yield tract-specific features. Please refer to the methods section for more details on the used datasets, preprocessing, parameter calculation, tract modeling and benchmark methods.

All classification and regression experiments described in the following were performed for each tract individually using a random forest with 100 trees, a maximum tree depth of 4 and no further hyperparameter optimization. To obtain reliable performance indicators, each experiment was realized as ten times differently seeded leave-one-out cross-validation (LOO-CV).

### RadTract features enable a rich representation of image information along tracts

Classic tractometry reduces the complete image information along the tract to a relatively small number of averages. For the two benchmark approaches, $n$ was set to 100, as suggested in the literature[47,48], resulting in 400 features extracted from the four parameter maps.

In contrast, RadTract leverages the well-established concept of radiomics for extracting as much information as possible from the image section covered by the respective tract of interest in the form of well-defined and standardized features[34,49]. To this end, RadTract subdivides each tract into $n$ parcels. For each parcel, we calculated a set of 18 first-order statistics, 14 shape-based, and 73 texture features, using RadTract, resulting in 105 features per parcel and parameter map. Since this process results in a very large number of features ($4 \cdot 105 \cdot n$), an automatic pre-selection of $k$ features was performed on the training set of each of the following ML experiments.

Details about the parcellation, feature calculation and the automatic feature pre-selection can be found in the methods section. Figure 1 visualizes the RadTract features for the CST. The complete list of features used in this work can be found in Supplementary Table 9.

### RadTract features enable improved automatic diagnoses

To analyze the value of RadTract features for automatic diagnosis, we performed classification experiments to identify patient subgroups with distinct diagnoses in all datasets. These subgroups not only comprised the classes "healthy" and "diseased" but different stages of the diseases, which made the task much more challenging but at the

same time more relevant. Please refer to the methods section for a description of the datasets and subgroups.

As a metric to quantify classification performance, we chose the area under the receiver operating characteristic (AUROC). We favor this metric over metrics such as the accuracy to avoid introducing a bias into the results due to arbitrary probability thresholds. In the multi-class case, we used the One-vs-the-Rest (OvR) strategy[50].

We optimized the number of selected features $k$ and, in case of RadTract, the number of parcels $n$ on the SCHZ dataset. The experiments on all other datasets were then performed using these optimized parameters.

In case of RadTract, $n$ was chosen for each tract individually to ensure roughly constant parcel thicknesses across tracts of different lengths: $n_v = n_{voxels}/v$, where $n_{voxels}$ is the average number of voxels traversed by the streamlines and $v$ is the desired thickness of the tract parcels in tract direction. $n_v$ was fixed for all subjects as the average value of ten random subjects.

Figure 2 shows AUROC scores for $k \in \{50, 100, 150, 200, 300, 400, 750, 1000, 1500\}$ of the two benchmark methods, as well as RadTract with four different parcellations ($n \in \{1, n_{2.5}, n_5, n_{7.5}\}$). For visualization purposes, only the mean values and the variance across repetitions and tracts are shown.

Table 1 shows the optimal values for $k$ and $n$, maximizing the mean and minimum AUROC while at the same time minimizing the variance that were used in all experiments. The resulting values for $n$ used in our experiments can be found in Supplementary Table 1.

RadTract features lead to improved results for all datasets and markedly outperformed classic tractometry (Centerline/Static) by 3.5/4.6, 4.6/6.5, 7.8/9.0, and 3.7/5.6 points AUROC on average across all tracts and repetitions in the datasets SCHZ, CAT, ADNI, and PPMI, respectively.

RadTract ranked first in 36 out of 46 tract/dataset combinations (14 statistically significant ($p < 0.05$), $p$-values can be found in Supplementary Tables 2–5), second in 6 and third in 4. The proposed

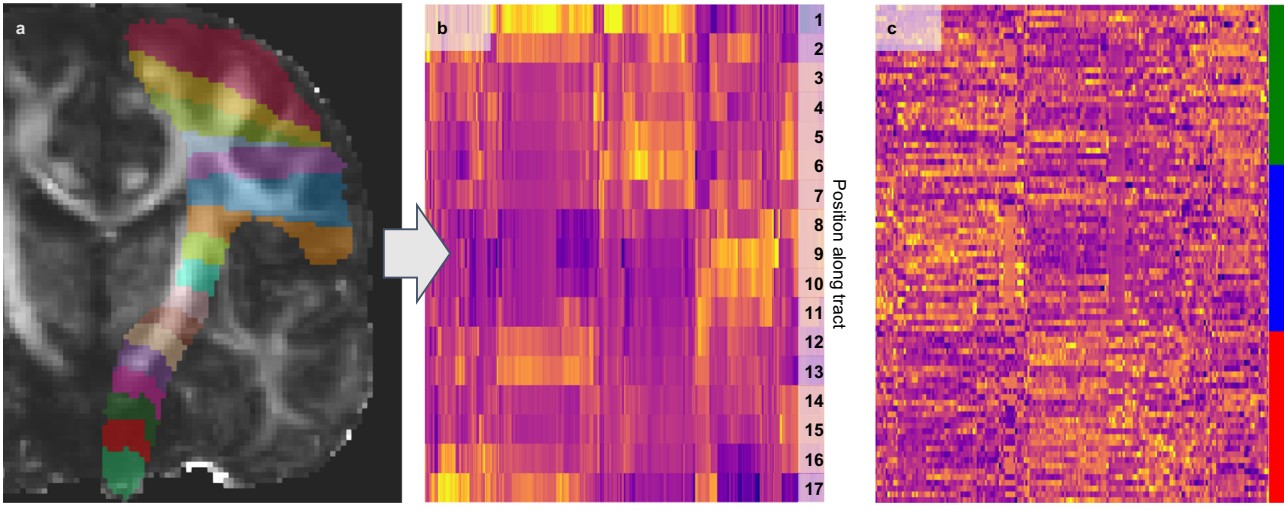

**Fig. 1 | Illustration of the RadTract process. a** RadTract-based CST parcellation of an exemplary subject. **b** Features corresponding to **a** as a heatmap. Per line, i.e., parcel, all 420 features are visualized. **c** The CST features (columns) of all CAT subjects (rows) with their respective class (color bar). Differences between the three classes are clearly visible in the features.

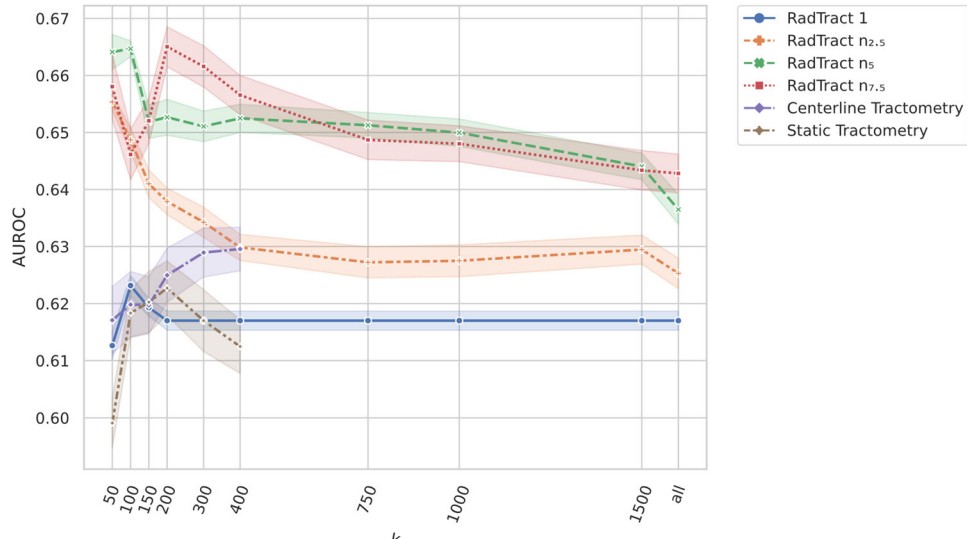

**Fig. 2 | AUROC results of the experiments performed to optimize the hyperparameters $k$ and $n$ for all successive experiments.** AUROC scores are presented as mean values +/− variance (14 tracts examined over 10 experiments each for each $k$). The optimal $k$ for each curve was determined as the $k$ that maximizes the product of minimum, maximum and 1/variance. See Table 1 for the optimal values of each approach.

approach further yields very promising results in 13 tracts, where tractometry features only yield results close to the level of random guessing (AUROC < 0.55). AUROC values > 0.7 are achieved in 5 tract/dataset combinations (RadTract) as compared to only 3 and 2 by the two benchmark approaches respectively. Figure 3 provides an overview over the classification performance of the applied approaches.

For all methods, intra-tract variations due to differently seeded repetitions are relatively low, with standard deviations of around 0.01. Results for all tracts individually can be found in Supplementary Figs. 1–8. A description of the statistical tests for significance can be found in the methods section.

### RadTract features enable improved prediction of demographic and clinical parameters

To demonstrate the potential of RadTract for tasks beyond automatic diagnosis and also beyond medical applications, we performed experiments to automatically predict demographic (age, number of pack-years, years of education) and clinical parameters (BPRS total, PANSS total, GAF scores, and olanzapine equivalents (OLZe)) on patients of the CAT dataset. A description of the individual parameters can be found in the methods.

Pearson's correlations between the predicted parameters and the true parameters are in general very low across all methods, except for the Age, as can be seen in Fig. 4a. At least weak correlations ($r > 0.15$) could be observed in 38, 29 and 27 tract/target combinations by the approaches RadTract, Centerline Tractometry and Static Tractometry, respectively.

Out of those, RadTract yielded the lowest mean squared error (MSE) in 31 tract/target combinations, while the benchmark approaches yielded the lowest MSE in only 11 and 5 combinations, respectively (see Fig. 4b). The results for all tracts and targets individually (MSE, Pearson's Correlation r, coefficient of determination $R^2$), as well as the ranges of the clinical parameters can be found in Supplementary Table 6 and Supplementary Figs. 9–29.

### RadTract enables a detailed analysis of feature, tract parcel and parameter map importance

RadTract allows a detailed analysis of the importance of different classes of features as well as of the different tract parcels, here demonstrated for the automatic diagnosis task. The importance of individual features is provided by the random forests in the form of the mean decrease in impurity introduced by each feature. Using the values of individual features, it is possible to determine the importance of complete feature classes or of the individual parameter maps as aggregates of the individual importance values.

The FA is by far the most important parameter for automated diagnoses (Fig. 5a). While this is the case in all datasets, it is particularly pronounced for SCHZ. In contrast, the RD parameter seems to hold the least information. While there are no large differences between the datasets for AD, ADC maps are relatively important for distinguishing CAT and ADNI subgroups and RD for distinguishing PPMI subgroups.

The top ten most important feature types, independent of the parameter map, are dominated by first order features (Fig. 5b). Three texture features can be found in the top-ten and no shape feature. The

#### Table 1 | Optimized parameters for *k* and *n* for RadTract and the two benchmark approaches

|  | n | k |
|---|---|---|
| RadTract | $n_5$ | 100 |
| Centerline Tractometry | 100 (fixed) | 400 |
| Static Tractometry | 100 (fixed) | 100 |

**a** Number of tracts per rank

|  | RadTract | Centerline Tractometry | Static Tractometry |
|---|---|---|---|
| Rank 1 | 36 (14) | 5 (0) | 5 (2) |
| Rank 2 | 6 | 25 | 15 |
| Rank 3 | 4 | 16 | 26 |

**b** Tracts where method ranked first

|  | RadTract | Centerline Tractometry | Static Tractometry |
|---|---|---|---|
| SCHZ | 10 (1) | 2 (0) | 2 (0) |
| CAT | 10 (4) | 3 (0) | 1 (1) |
| ADNI | 10 (6) | 0 (0) | 0 (0) |
| PPMI | 6 (3) | 0 (0) | 2 (1) |

**c** Not better than random

| RadTract | Centerline Tractometry | Static Tractometry |
|---|---|---|
| 4 | 17 | 20 |

**d** Per-dataset AUROC

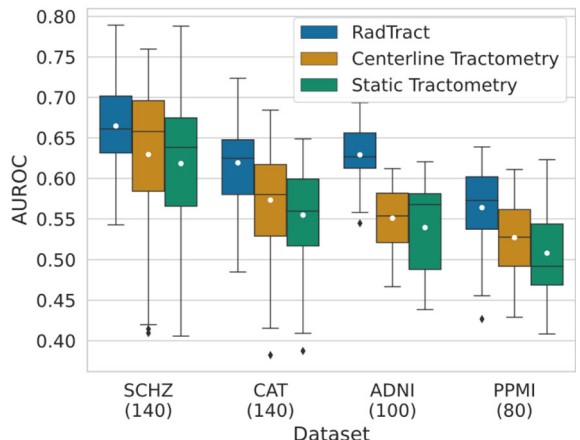

**Fig. 3 | Classification results. a** Number of tracts in which the compared approaches were ranked first, second or third, respectively. **b** Number of tracts and in which dataset the compared approaches ranked first. **c** Number of tracts across datasets where the compared approaches performed not better than random, i.e., AUROC < 0.55. Numbers in brackets in **a**–**c** indicate the number of statistically significant results ($p < 0.05$) obtained using Delong's method without correction for multiple comparison. P-values can be found in Supplementary Tables 2–5.

Details about the statistical analysis can be found in the methods section.
**d** AUROCs of all methods on all datasets across tracts and repetitions, i.e., the summarized classification results across tracts. Results for all tracts individually can be found in Supplementary Figs. 1–8. We use standard box plots including the mean value (white marker) with 1.5 IQR whiskers. The number of experiments displayed in **d** is indicated in brackets in the graph legend.

most important features for each dataset individually can be found in Supplementary Figs. 30–33.

When looking at complete feature classes (first order, texture and shape), further differences become apparent. Aggregated, the texture features are by far the most important features (Fig. 5c). On average, individual first order features are more important (Fig. 5d). This can be attributed to the fact that much more texture features exist, and while each of them only contributes a little to the overall classification result, in total they are quite important. An exception is the ADNI dataset, where individual texture features are as important as first order features. In summary, texture features seem to be the most relevant feature class in our experiments, followed by first order features.

By analyzing the aggregated feature importance values for individual tract parcels, it is possible to assess the relevance of certain tract locations for distinguishing the patient subgroups. Supplementary Figs. 34–79 provide parcel importances for all tracts, datasets and methods.

## Discussion

We present radiomic tractometry (RadTract), a new approach for quantifying fibrous tissue such as the brain WM along its course. RadTract extends the state-of-the-art in tract-specific tissue analysis (tractometry) from simple tract profiles to descriptive feature sets that capture the full richness of the image information along the tract and enable improved predictive modeling. This is achieved by computing 105 standardized first-order, shape, and texture features per parcel and parameter map in contrast to the limited information provided by classical approaches (tractometry). To our knowledge, this is the first approach to translate the concept of radiomics, which has been successfully used in many other radiological domains, to the world of tract-specific WM analysis.

We conducted a series of experiments in multiple psychiatric and neurological datasets, illustrating the general applicability of RadTract as well as its promising performance compared to the state-of-the-art in various tasks. Our experiments show that RadTract is capable of extracting much more meaningful information from images than it is possible with classic tractometry, enabling new insights even when using well-studied and long-established parameter maps such as the FA, ADC, AD and RD. In general, RadTract supports arbitrary parameter maps besides the ones chosen in this work as well as other image

contrasts as input, which, if chosen smartly for the respective task at hand, are expected to yield even more valuable features.

The improvements using RadTract are most pronounced in the ADNI dataset. Interestingly, this is also the dataset where texture features show a much higher importance compared to the other datasets. This could for example be related to the nature or severity of the specific pathology, but a thorough investigation of this aspect is beyond the scope of this work and planned for future projects.

While our experiments showed that RadTract outperforms the state-of-the-art in many cases, overall classification performance can still not be considered sufficient for reliable subject-level predictions. This is also the case for the performed regression experiments, where correlations are mostly close to zero across methods. Nevertheless, RadTract yields lower errors in more tracts that hold at least some information that might be suitable for group analysis.

A limitation of RadTract related to this aspect is its large feature set. On the one hand, this is its greatest asset. On the other hand, it makes the issue of dataset size increasingly critical and studies involving larger samples may be required to leverage RadTract's full potential by enabling a more reliable selection of robustly generalizing features from the complete feature set. In this context, automatic feature selection plays a crucial role. While we obtained promising results with a simple univariate feature selection approach, using more advanced techniques could further improve the performance of Rad-Tract features in the downstream tasks. Powerful feature selection in combination with large sample sizes gain even more relevance when considering even further increased sets of features, e.g., by including features from filtered versions of the original image, which has the potential to further boost predictive performance[51]. While we show increased classification performance using RadTract, these results are only statistically significant in a part of the analyzed tracts. Larger cross-sectional and longitudinal datasets of different patient populations will be required to obtain a more comprehensive picture of RadTracts performance in the other tracts.

An approach that has to be discussed in the context of our work is the Detect system recently presented by Chamberland and colleagues[52]. As RadTract, Detect has the goal of improving patient-level predictions using tract-specific features. Nevertheless, Detect does not aim at improving the used features. In fact, it uses standard

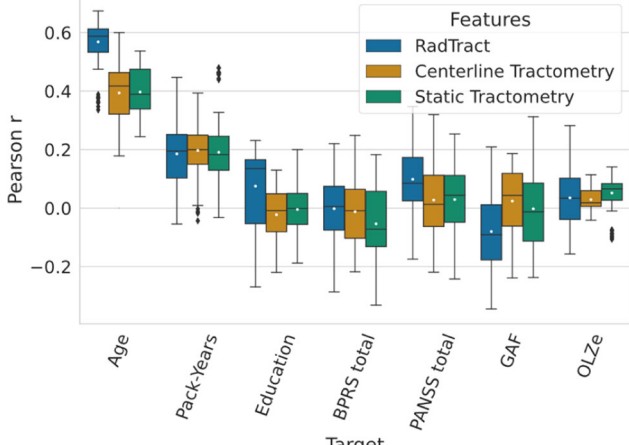

**a** Pearson correlation per target

**b** Tracts with lowest MSE and r > 0.15

| Measure | RadTract | Centerline Tractometry | Static Tractometry |
|---|---|---|---|
| Age | 12 | 2 | 0 |
| Pack-Years | 5 | 6 | 2 |
| Education | 5 | 0 | 1 |
| BPRS total | 2 | 1 | 0 |
| PANSS total | 4 | 2 | 1 |
| GAF | 1 | 0 | 1 |
| OLZe | 2 | 0 | 0 |

**Fig. 4 | Regression results. a** Pearson's correlations for three demographic (Age, Pack-Years, Education) and four clinical (BPRS total, PANSS total, GAF, OLZe) parameters. We use standard box plots including the mean value (white marker) with 1.5 IQR whiskers. The number of experiments displayed in **a** is 140 (14 tracts examined over 10 experiments each). Correlations are low for all methods in most targets. **b** Tract/target combinations where correlations are at least somewhat apparent (*r* > 0.15). Here, RadTract yields the lowest MSE more often than the benchmark approaches. The results for all tracts and targets individually can be found in Supplementary Figs. 9–29.

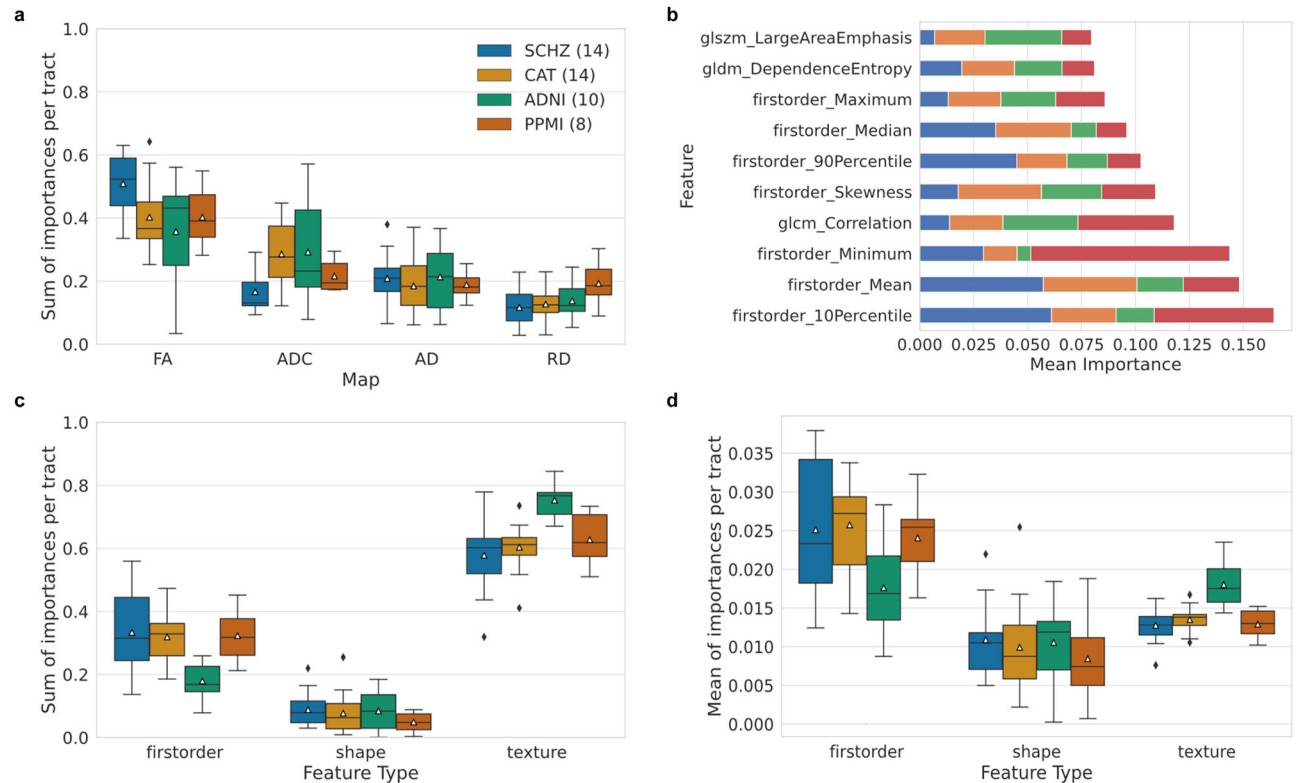

**Fig. 5 | RadTract enables a detailed analysis of feature, tract parcel and parameter map importance. a** Importance of the individual parameter maps. **b** Top ten most important features across datasets. **c**, **d** Importance of individual feature classes (first order, shape and texture). Texture features have a relatively low mean importance (**d**) but due to their large number a high aggregated importance (**c**) and therefore most likely still a large influence on the result. We use standard box plots including the mean value (white marker) with 1.5 IQR whiskers. The number of tracts in **a**, **c** and **d** is indicated in the graph legends.

Centerline Tractometry features. The important aspect of Detect is that it is based on an unsupervised approach for outlier detection, making it a powerful tool for detecting tissue changes without being trained on them. This makes Detect complementary to RadTract in the sense that Detect could very well employ RadTract features instead of classic tractometry features, which has the potential to increase its performance, as indicated by our results. We are planning to investigate this in future studies.

To be considered when using RadTract and tractometry in general is its dependence on the accurate delineation of the target tract, and segmentations that for example spill into neighboring gyri might confound the resulting features. When performing studies using RadTract, it is therefore advisable to perform a proper quality check of all used segmentations. A related aspect is the actual definition of the tract shapes and courses, which might vary between segmentation methods. RadTract is agnostic towards the tool used to generate the tracts, and in this proof of concept study the most important aspect to ensure is that all methods use the same tract definitions as input. Since it is a popular approach that covers all tracts of interest for our work, we used TractSeg. This choice might not be ideal for other studies, though, and the optimal tool for the task at hand has to be chosen each time anew.

In summary, RadTract defines a new state-of-the-art for tract-specific tissue analysis. We expect the presented work to be a starting point for a new generation of imaging biomarkers in the neuroscientific research domain and beyond. We believe that, used as an out-of-the-box tool for the calculation of advanced and standardized tract-specific imaging features, RadTract will be a valuable resource for the research community, opening up new research avenues and stimulating new investigations of the human brain white matter.

## Methods

In the following subsections, we will describe the data used in our experiments, the preprocessing of the data, the used benchmark approaches, the actual RadTract methodology itself, consisting of the tract parcellation and the feature calculation, as well as the employed statistical tests and some implementation details of RadTract.

### Datasets

We included dMRI data of 216 subjects from the ADNI dataset (all ADNI phases) in our classification experiments. The data comprises three subgroups: 72 patients with diagnosed Alzheimer's disease, 72 patients with mild cognitive impairment, and 72 healthy controls. The three groups were matched for age and sex (see Supplementary Table 10). All datasets as well as further information about the data are accessible via https://ida.loni.usc.edu/. The ADNI was launched in 2003 as a public-private partnership, led by Principal Investigator Michael W. Weiner, MD. The primary goal of ADNI has been to test whether serial magnetic resonance imaging, other biological markers, and clinical and neuropsychological assessment can be combined to measure the progression of mild cognitive impairment and early Alzheimer's disease. For up-to-date information, see www.adni-info.org. Data were acquired on GE and Siemens 3T MRI scanners with a varying number of gradient directions between 16 and 48 at $b = 1,000 s/mm^{-2}$ and a varying anisotropic resolution between 1.0 and 2.7 mm. The IDs of all included subjects can be found in Supplementary Table 7 and the corresponding imaging parameters can be accessed via the dataset webpage.

We included dMRI data of 129 subjects from the PPMI dataset (baseline visit) in our classification experiments. The data comprises three subgroups: 43 patients with diagnosed Parkinson's disease, 43 patients with prodromal Parkinson's disease, and 43 healthy controls.

The three groups were matched for age and sex (see Supplementary Table 10). All datasets as well as further information about the data are accessible via https://ida.loni.usc.edu/. Data were acquired on a Siemens 3T MRI scanner with 64 directions with varying b-values of $b = 600 s/mm^{-2}$ and $b = 1,000 s/mm^{-2}$ as well as 2 mm isotropic resolution. The IDs of all included subjects can be found in Supplementary Table 7 and the corresponding imaging parameters can be accessed via the dataset webpage.

We included dMRI data of 98 subjects from the SCHZ dataset in our classification experiments. The data comprises two subgroups: 49 patients with diagnosed schizophrenia and 49 healthy controls. The groups were matched for age and sex (see Supplementary Table 10). This data was obtained from the OpenfMRI database (https://openfmri.org/dataset/ds000030/)[46]. Its accession number is ds000030. Data were acquired on a Siemens 3T Tim Trio MRI scanner with 64 directions at $b = 1,000 s/mm^{-2}$ and 2 mm isotropic resolution. The IDs of all included subjects can be found in Supplementary Table 7 and the corresponding imaging parameters can be accessed via the dataset webpage.

We included dMRI data of 87 subjects from the CAT dataset in our classification experiments. The data comprises three subgroups: 30 schizophrenia patients with catatonia, 29 schizophrenia patients without catatonia, and 28 healthy controls. The groups were matched for age and sex (see Supplementary Table 10). For the regression experiments, we used 59 schizophrenia patients from the CAT dataset and 49 additional schizophrenia patients which were not previously considered in the CAT analyses due to the lack of matching with healthy controls (108 patients total). The following demographic and clinical measures were included in our experiments:

- Age in years
- Pack-Years: the number of packs of cigarettes smoked per day by the number of years the person has smoked.
- Education: the number of years the person spent in an educational institution, such as high school or university.
- BPRS total: aggregated score on the Brief Psychiatric Rating Scale (BPRS), measuring the severity of various psychiatric symptoms.
- PANSS total: aggregated score on the Positive and Negative Syndrome Scale (PANSS), measuring symptom severity of patients with schizophrenia.
- GAF: score on the Global Assessment of Functioning scale, measuring the social, occupational, and psychological functioning of the person.
- OLZe: indicating the daily doses of antipsychotic medication in terms of Olanzapine equivalents (OLZe).

Data were acquired at CIMH on a Siemens 3T Tim Trio MRI scanner with 60 directions at $b = 1,500 s/mm^{-2}$ and 1.7 mm isotropic resolution. The studies on the acquisition of CAT and healthy control data were approved by the local ethics committees (Medical Faculties Mannheim and Heidelberg at Heidelberg University, Germany). Written informed consent was obtained from all participants after a detailed explanation of the aims and procedures of the study. The CAT participants received financial compensation for their participation in the study. Further details about the dataset can be found in the work presented by Wasserthal and colleagues[27].

## Data preprocessing and tract modeling
The following artifact and noise correction steps were conducted for all dMRI images using MRtrix and FSL[53,54]: noise level estimation and denoising[55], Gibbs ringing removal[56], eddy current and inhomogeneity distortion correction[57,58] as well as bias field correction[59]. The corrected images were then rigidly registered and resampled to the MNI-space FA template (1.25 mm isotropic resolution) contained in the TractSeg

package using MITK Diffusion. Brain masks were calculated using FSL Bet[60]. Tensors and FA maps were calculated and constrained spherical deconvolution (CSD) with a successive extraction of the three principal fiber directions (peaks) was performed using MRtrix[61]. The peaks served as input to TractSeg, which was used to calculate tract segmentations, tract start- and end-region segmentations, tract orientation maps, and as well as tractograms of each tract. A complete list of the used commands and parameters can be found in Supplementary Table 8.

## Analyzed tracts
The corpus callosum (CC) serves as one of the most prominent tracts in the human brain that is responsible for the mediation of inter-hemispheric transfer, in terms of increased inhibition or reduced facilitation[62]. Intact transcallosal functioning is essential for sustained attention, motor control, and synchronization of bilateral movements. For these reasons, the CC plays a crucial role in the pathophysiology of all psychiatric disorders. Therefore, the Rostrum (CC_1), Genu (CC_2), Rostral Body (CC_3), Anterior Midbody (CC_4), Posterior Midbody (CC_5) and Isthmus (CC_6) of the CC were analyzed in all datasets. Due to frequent errors in the TractSeg results in all datasets, the Splenium of the CC (CC_7) was excluded from the analysis (see Supplementary Fig. 80). Because this manuscript aims to present a new method rather than the pathophysiology of each of the four psychiatric cohorts, besides CC, we have focused on the pathophysiologically most plausible WM tracts. The choice of these additional WM tracts per cohort will be described in the following paragraphs.

The scientific community is becoming more interested in cerebellar circuitry as a result of the cerebellum's crucial involvement in motor, cognitive, and emotional activities as well as the deterioration of its functioning with age[63]. Further, recent neuroimaging studies showed that the cerebellum is involved in Alzheimer's disease[64]. Since WM microstructural alterations of the cerebellum are relevant for both Alzheimer's disease and mild cognitive impairment patients, we decided to include the left and right Inferior Cerebellar Peduncle (ICP) and Superior Cerebellar Peduncle (SCP) in our experiments on the ADNI dataset.

WM microstructural alterations of the corticospinal tract (CST) are discussed as possible biomarkers of Parkinson's disease[65,66]. Furthermore, previous studies also showed that WM microstructural alterations of CST could serve as an early marker for prodromal Parkinson's disease[67–69]. Since CST plays a crucial role in all three motor stages of Parkinson's disease (e.g., silent, prodromal and clinical Parkinson's disease), we decided to include the left and right CST in our experiments on the PPMI dataset.

SCHZ patients show structural and functional alterations in both cortical (e.g. frontal, prefrontal, parietal, temporal, cingulate, and insular cortex)[70,71] and subcortical (e.g. hippocampus, amygdala, nucleus accumbens, striatum, and thalamus) regions[72]. Therefore, besides CC, we included WM tracts connecting the majority of the neurobiologically plausible cortical and subcortical regions, i.e., left and right Thalamo-Prefrontal Tract (T_PREF), Thalamo-Parietal Tract (T_PAR), Striato-Parietal Tract (ST_PAR) and Striato-Fronto-Orbital Tract (ST_FO), in our experiments on the SCHZ dataset.

The majority of MRI studies proposed a pathophysiological model of catatonia including right hemispheric neural network abnormalities that include the medial and lateral orbitofrontal cortex, prefrontal cortex, supplementary motor area, primary motor cortex, posterior parietal cortex, anterior cingulate cortex, amygdala, thalamus, and cerebellum, respectively[73–75]. Therefore, besides CC, we included WM tracts connecting these regions, i.e. the left and right CST, Striato-Fronto-Orbital Tract (ST_FO), Thalamo-Premotor Tract (T_PREM) and Striato-Premotor Tract (ST_PREM), in our experiments on the CAT dataset.

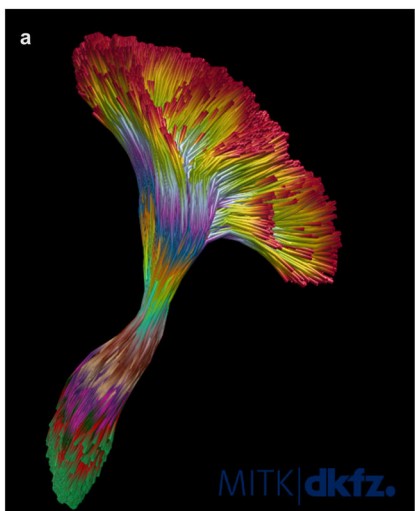
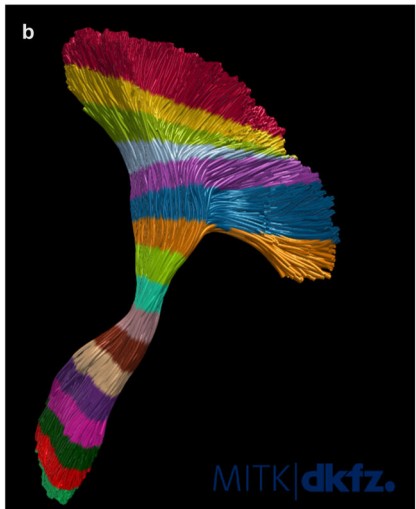
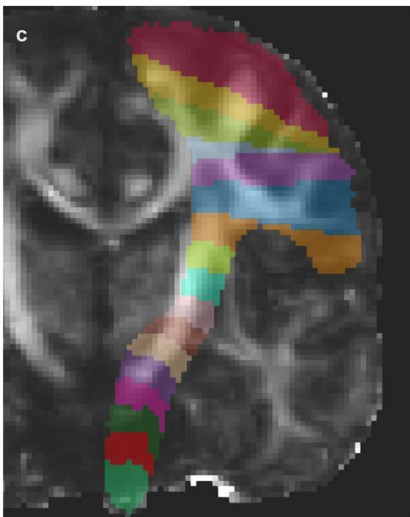

**Fig. 6 | Comparison of CST parcellations. a** Static Tractometry. **b** Centerline Tractometry. **c** RadTract. The RadTract parcellation corresponds to the Centerline Tractometry parcellation, only in voxel space instead of streamline space. For illustration purposes, all approaches were parameterized with the same number of parcels $n$ instead of the $n$ used in the experiments.

## Benchmark methods

As described in the introduction, tractometry involves evaluating the parameter map of interest along the points of the individual fibers. Each value is assigned to one of $n$ parcels depending on its position. The values within each parcel are then aggregated, usually by averaging, resulting in a vector of scalar values along the tract that can be used for further analysis. There are two main approaches for parcel assignment:

Approach (1) statically resamples the streamlines to $n$ points (1) and the parcel ID for each point is directly given by the point's position along the streamline[23]. Here, we refer to this approach Static Tractometry.

Approach (2) assigns each value at a streamline point to the closest point on a tract-centerline composed of $n$ points (2)[48]. In this case, the parcel ID is given by the position of the respective closest point on the centerline. This approach avoids parcel assignment errors of approach (1), which arise from misalignment among the individual streamlines, causing image values at the same spatial position to be assigned to different parcels based on their respective positions along the streamline. Here, we refer to this approach as Centerline Tractometry.

Both approaches were used as benchmark methods for our proposed approach in all experiments. As described in the results, $n$ was set to 100, as suggested in the literature[47,48]. Figures 6a and b illustrate both parcellation types.

## Tract parcellation

As described in the previous section, classic tractometry assigns parcels to points in streamline space. RadTract, on the other hand, calculates radiomics features in the image- or voxel-space. To this end, RadTract assigns each voxel of the corresponding binary tract envelope to a parcel using the same centerline-based approach as used in Centerline Tractometry. The binary envelope of a tract is created by labeling each voxel of the corresponding image that is traversed by a fiber with 1 and all other voxels with 0. Small holes arising from sparsely populated tracts are filled using a standard morphological closing operation on the binary mask. Figure 6c illustrates the resulting Rad-Tract parcellation in voxel space. In case of the voxel-space parcellation of RadTract, $n = 100$, as used for the benchmark approaches, is not feasible since the voxel spacing imposes a natural and tract-dependent upper limit on $n$. The choice of $n$ for RadTract is described in the results section.

## Feature calculation

RadTract calculates 105 radiomics features in each parcel and for each parameter map. The calculated features can be categorized in the following types:

- First Order Statistics (18)
- Shape-based (14)
- Texture (73):
  - Gray Level Cooccurence Matrix (22)
  - Gray Level Run Length Matrix (16)
  - Gray Level Size Zone Matrix (16)
  - Neighbouring Gray Tone Difference Matrix (5)
  - Gray Level Dependence Matrix (14)

Feature calculation is based on pyradiomics, a widely used open-source Python package for radiomics feature calculation[34]. The Feature calculation can be easily customized using yaml-based parameter files. The complete list of features used in this work can be found in Supplementary Table 9 and the yaml file used to parameterize the feature extraction is included in the supplementary code.

Since RadTract produces numerous features per tact, automatic feature selection is vital for later analysis. A large variety of feature selection techniques exist, and the best choice depends on the concrete task. A detailed analysis of this aspect would go beyond the scope of this work. In our experiments, we first removed all constant features as well as all highly correlated (Pearson correlation > 0.95) and therefore likely redundant features. Second, we used a simple and fast univariate feature selection implemented in scikit-learn on the training data to further automatically reduce the respective input feature set to $k$ features[76]. This type of feature selection works by selecting the best features based on univariate statistical tests. In case of a classification experiments, this method computes the ANOVA F-value between each feature and the target variable. In case of a regression task, this is done by calculating the linear correlation of each feature with the target variable. Both approaches return the F-value and p-value for each feature. The F-value measures the difference in means between the classes or the linear correlation coefficient, for classification and regression respectively, while the p-value measures the significance of the difference/correlation. The higher the F-value and the lower the p-value, the more significant the feature is in predicting the target variable. This approach for feature selection is supervised, meaning it does indeed use the target variables. Therefore, the feature selection is performed on the training samples only, not on the test samples. The

process is deterministic, meaning that given the same samples, the same features will be selected. Nevertheless, the selected features might be different across folds.

## Statistical tests

Tests for significance of the classification experiments were performed using Delong's method for statistical comparisons of ROC curves (https://github.com/yandexdataschool/roc_comparison, commit hash 44fcd23). A tract-level-result was deemed significant if the mean p across repetitions was smaller than 0.05. In case of a multi class experiment, the tests were performed for each class independently using the One-vs-the-Rest (OvR) strategy and an improvement was deemed significant if $p < 0.05$ for at least one of the classes.

## Implementation details

We used the support vector classification as well as random forest classification and regression implemented in scikit-learn (v1.1.2) in our implementation of the RadTract parcellation function as well as all classification and regression experiments[76]. Default parameterization was used if not stated otherwise. Pyradiomics v3.0.1 was used for all radiomics feature calculations[34]. Further used python packages include numpy (v1.23.3), scipy (v1.9.1), pydicom (v2.3.0), nibabel (v4.0.2), skimage (v0.19.3), dipy (v1.5.0), TractSeg (v2.7) and vtk (v9.2.0)[47,77–82]. Python version 3.10 was used in all experiments.

## Reporting summary

Further information on research design is available in the Nature Portfolio Reporting Summary linked to this article.

## Data availability

In this study, no new data was gathered. Instead, it incorporated four existing datasets. The first two, the ADNI and PPMI datasets, are accessible through the Image and Data Archive (IDA, https://ida.loni.usc.edu/). For the ADNI data, access can be obtained through the IDA's ADNI section (https://adni.loni.usc.edu/data-samples/access-data/), and for the PPMI data, access can be obtained through the PPMI's information page (https://www.ppmi-info.org/access-data-specimens/download-data). The third dataset, SCHZ, is available in the Open-Neuro database under the accession code ds000030 (https://openneuro.org/datasets/ds000030/versions/00016/download). The fourth dataset (CAT) is a non-public dataset acquired at the Central Institute of Mental Health (CIMH). Due to data privacy laws, the CAT data are protected and not available. The IDs of all subjects included from the public datasets can be found in Supplementary Table 7, and the corresponding imaging parameters can be accessed via the dataset webpage. Source data are provided with this paper.

## Code availability

The RadTract code is available as supplementary software. RadTract version 0.1.9 was used for the presented experiments[83]. Updated versions can be found on https://github.com/mic-dkfz/radtract and https://pypi.org/project/radtract/.

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

## Acknowledgements

This work was supported by the German Research Foundation (DFG) grant numbers MA6340/10-1, MA6340/12-1, and HI1928/5-1.

We further want to thank the division of biostatistics of the German Cancer Research Center (DKFZ) for their support.

Data of healthy subjects in the CAT dataset was provided by Prof. Dr. Dr. Heike Tost (German Federal Ministry of Education and Research (BMBF) grant 01GQ1102) and Dr. med. Lena Geiger-Primo. We would also like to thank Prof. Dr. med. Andreas Meyer-Lindenberg for his support in recruiting patients and healthy subjects in the CAT dataset.

Data used in preparation of this article were obtained from the Alzheimer's Disease Neuroimaging Initiative (ADNI) database (adni.loni.usc.edu). As such, the investigators within the ADNI contributed to the design and implementation of ADNI and/or provided data but did not participate in analysis or writing of this report. A complete listing of ADNI investigators can be found at: http://adni.loni.usc.edu/wp-content/uploads/how_to_apply/ADNI_Acknowledgement_List.pdf. Data collection and sharing for this project was funded by the Alzheimer's Disease Neuroimaging Initiative (ADNI) (National Institutes of Health Grant U01 AG024904) and DOD ADNI (Department of Defense award number W81XWH-12-2-0012). ADNI is funded by the National Institute on Aging, the National Institute of Biomedical Imaging and Bioengineering, and through generous contributions from the following: AbbVie, Alzheimer's Association; Alzheimer's Drug Discovery Foundation; Araclon Biotech; BioClinica, Inc.; Biogen; Bristol-Myers Squibb Company; CereSpir, Inc.; Cogstate; Eisai Inc.; Elan Pharmaceuticals, Inc.; Eli Lilly and Company; EuroImmun; F. Hoffmann-La Roche Ltd and its affiliated company Genentech, Inc.; Fujirebio; GE Healthcare; IXICO Ltd.; Janssen Alzheimer Immunotherapy Research & Development, LLC.; Johnson & Johnson Pharmaceutical Research & Development LLC.; Lumosity; Lundbeck; Merck & Co., Inc.; Meso Scale Diagnostics, LLC.; NeuroRx Research; Neurotrack Technologies; Novartis Pharmaceuticals Corporation; Pfizer Inc.; Piramal Imaging; Servier; Takeda Pharmaceutical Company; and Transition Therapeutics. The Canadian Institutes of Health Research is providing funds to support ADNI clinical sites in Canada. Private sector contributions are facilitated by the Foundation for the National Institutes of Health (www.fnih.org). The grantee organization is the Northern California Institute for Research and Education, and the study is coordinated by the Alzheimer's Therapeutic Research Institute at the University of Southern California. ADNI data are disseminated by the Laboratory for Neuro Imaging at the University of Southern California

Data used in the preparation of this article were obtained [on 7, 20 2022] from the Parkinson's Progression Markers Initiative (PPMI) database (www.ppmi-info.org/access-data-specimens/download-data), RRID:SCR_006431. PPMI – a public-private partnership – is funded by the Michael J. Fox Foundation for Parkinson's Research funding partners 4D Pharma, Abbvie, Acurex Therapeutics, Allergan, Amathus Therapeutics, ASAP, Avid Radiopharmaceuticals, Bial Biotech, Biogen, BioLegend, Bristol-Myers Squibb, BlueRock Therapeutics, Calico, Celgene, Dacapo Brain Science, Denali, The Edmond J. Safra Foundation, GE Healthcare, Genentech, GlaxoSmithKline, Golub Capital, Handl Therapeutics, Insitro, Janssen Neuroscience, Lilly, Lundbeck, Merck, Meso Scale Discovery, Neurocrine Biosciences, Pfizer, Piramal, Prevail, Roche, Sanofi Genzyme, Servier, Takeda, Teva, UCB, Verily, and Voyager Therapeutics.

## Author contributions

P.N., D.H., and K.M.-H. designed the study. P.N. performed the data analysis. D.H. collected and curated data. P.N., D.H., K.M.-H. interpreted the results, wrote, and revised the manuscript.

## Funding

## Competing interests

The authors declare no competing interests.
