## [Peer Review File · Nature Communications]

Radiomic tractometry reveals tract-specific imaging biomarkers in white matterREVIEWER COMMENTS

Reviewer #1 (Remarks to the Author):

This paper introduces two separate innovations to the analysis of brain white matter tracts from in vivo diffusion MRI data. The first is in how white matter tracts are sampled for the purpose of analysis. The other is in the use of a set of radiomic features for the purpose of machine learning analysis. Among its strengths is the fact that the paper includes analysis of several datasets on a range of different topics, showing potentially impressive improvements in performance. In addition, the paper includes an open-source software implementation of its main components. This should make the work more reproducible and extensible in the hands of others.

However, the paper also has some weaknesses. My main concern is that the results presented do not attempt to distinguish between the contributions of these two innovations. That is, it is not clear what are the relative contributions to improvements in performance from the use of radiomics features, and those that are due to the new parcellation scheme that is proposed. For example, one wonders how these algorithms would perform if the new parcellation scheme was used to produce FA profiles of the kind that is presented in Figure 1c and these profiles were used as features in the ML tasks presented in the Results section. The supplementary results suggest that at least in some cases, the parcellation by itself produces substantial improvements, and that the use of radiomics can help with additional improvement, but is by no means a dominant part of the improvement. Despite the inclusion of these in the supplement, I would like to see this question elevated with a more systematic analysis in the main results.

Figure 1 and the Introduction section introduce important issues with two different existing methods for white matter tract parcellation (referred to as Method 1 and Method 2 in lines 74-76). However, the empirical comparisons in the results focus only on one of these (Method 2), leaving me unconvinced that the results are as damning as suggested to previous methods. To establish the novelty of this approach, it would be important to show that the differences are similarly large with method 1, which should more closely resemble the parcellation approach introduced here (e.g., in terms of Dice coefficients around the

fanning parts of the bundles).

I also have some concerns regarding some of the statistical methods that were used:

1. Given the dimensions of the feature-set that is generated for every individual in every tract (e.g., 17 times 1,106 in the example shown in Figure 3) there must be a pretty large degree of redundancy. The authors mention that they performed univariate feature selection down to 500 using scikit learn methods, but it is not clear how this was done. What does this feature selection do? Does it use the target categories for selection? Which of the thousands of features entered into the model are retained? Is it always a particular set of radiomic features? Specific locations in each tract?

2. The authors use a non-parametric test for comparisons of ROC curve results over ten rounds of LOOCV. To the best of my knowledge this approach violates the assumptions of the test. This is due to the fact that the 10 AUROC numbers obtained in these 10 experiments are not independent samples, because there is a high degree of overlap between each of the samples. The authors could instead use Delong's method for statistical comparisons of ROC curves. Fortunately, there is an open-source implementation of this algorithm here: https://github.com/yandexdataschool/roc_comparison.

3. A similar problem occurs for the Pearson correlation comparisons, where each of the 12,470 regression experiments is considered an independent experiment in the Wilcoxon test used, but these are highly dependent, because they are derived from repeated resampling of the same original sample.

4. Another fact that bears explaining is how individual tracts seem to be able to predict better than the entire dataset. For example, it seems that parts of the CC make better classification prediction for the PPMI dataset than the entire dataset does.

Finally, some comments about the framing of the paper, which could be addressed through revisions to the Introduction/Discussion: The paper introduces ideas from the nascent field of radiomics into tractometry. This is a very clever translation of recently-developed

methods into a brand new application. However, I think that it would be worth explaining a bit more about (1) what radiomics is; I think that it is not a concept that has pervaded the MRI literature sufficiently yet to justify the implicit assumption that the target audience of this article has already heard about these methods (this reviewer, for one, had not heard about these methods before reading this manuscript), and (2) to explain what aspects of radiomics were used here. In addition, I believe that the paper fails to address recent tractometry findings that do allow comparisons of individual participants and makes a different use of the information in the bundles. These findings were introduced in a paper by Chamberland et al. in Nature Computational Science volume 1, pages 598–606, in a method called "Detect". That method -- which does rely on "centerline" methods -- seems better suited to address the issue raised in lines 64-66 of the Introduction. A comparison of the new methods to Detect (at least at the conceptual level) would be important to establish some of the claims made in this paper. I think that another limitation that the authors need to acknowledge is that they limited their analysis to fractional anisotropy and there may be important information in other image statistics.

Minor comments:

- Many of the Figures include font sizes that render the annotations completely illegible.
- Error bars used in Figures (e.g., in Figure 4b) should be explained. Are those standard boxplots?
- How were results chosen to be presented in Figure 4c? The choice of bundles across datasets seems inconsistent.

Reviewer #2 (Remarks to the Author):

Let me start by saying that bringing radiomics features in the world of tractometry is a great idea. Results on ADNI, CAT, PPMI, SCHZ seem promising and an improvement over current state-of-the-art tractometry. I do agree with authors that RadTract opens new potential for white matter analytics, especially given that this is also packaged as an open-source/open-science package.

Having said this, I have a few concerns with the current state of the manuscript that attenuate my enthusiasm. The paper is currently more a proof-of-principle than a final “product”.

Major concern:

Diffusion MRI tractometry already suffers from the curse of dimensionality, i.e. N tracks, M metrics, R track-parcels, already leading to a massive set of $N \times M \times R$ features. Here, on top of that, radiomics features are added, 1106 radiomic features for each track, each metrics, each track-parcels. This is a monstrous number. At the moment, to account for this, authors use

- an automatic feature selection to only select 500,
- use the FA metric only in the experiments and results,
- use a subset of tracks of interest in the different databases

Justification for 500, FA-only, and subset of tracks per database are poor, especially on the different diseases (e.g. AD, PD, ...). You have selected mostly CC-related tracks. For me, authors have not figured out a RadTract “product” that would deal with ALL the data appropriately and have data-driven dimensionality reduction techniques implemented. RadTract is a nice prototype for future in-death tractometry research.

Minor Concerns:

- 1) The new track parcellation approach is great. It seems to work. But for me, it deviates the high-level message of this work. Especially, in a Nature Comm publication. With the track parcellation included, the paper reads like a technical paper, with an incremental improvement in track sub-divisions. I would suggest publishing this part of the work in a methods paper or put less emphasize on it and send this part to supp. materials.
- 2) Because of 1) above, I don't find the paper particularly well-written or visually appealing.
- 3) As pointed above, if this paper is to be accepted, authors need to show how to include most dMRI metrics or DTI metrics in the pipeline, not just FA.
- 4) Even with the new track parcellation, there are clearly some tracks that split in 2 or 3, where the track-parcellations, e.g. CC3, CC5 in the supp materials. Parts of the tracks, near the endpoints seem to clearly finish in different gyri. To me, this is a clear limitation of

tractometry that hasn't be solved here. It is a multi-scale one. Should the track of interest be better defined? For example, defined at a scale where the endpoints cannot span more than one gyrus? This is an open question. But, somehow, I feel authors are ignoring it now. It should be added in a discussion.

Overall, great work. I'm being critical but I very much like the work.

Reviewer #3 (Remarks to the Author):

My experience is mainly in radiomic analysis of conventional MR images and my comments will therefore focus on this aspect. The aim of the proposed paper is to demonstrate, using 4 cohorts of moderate size, the added value of radiomic analysis performed on sub-segments of white matter bundle trajectories compared with conventional tractometry analysis. To this end, a new parcellation method based on an SVM technique is proposed, and compared with a conventional method based on an centroid-assignment strategy. From this new classification, sub-segments are extracted that serve as regions of interest for radiomic analysis. The analysis is original and the number of articles on the subject is limited.

However, the analysis contains a number of grey areas that need to be clarified.

- The abstract is very macroscopic and does not at all highlight the originality of the proposed work or the methodology implemented.
- In the introduction, only one of the assignment methods is properly detailed and illustrated (Figure 1), even though it is an important step in the paper. Assignment method number 1 deserves more detail. Moreover, it is not clear to me in the rest of the paper why only one assignment method (centroid-based) is used as a benchmark. Figure 1c is also completely incomprehensible as it stands.
- The number of parcels is automatically calculated per tract so as to obtain a number of voxels equal to 5 per parcel. This choice was made arbitrarily, but raises several questions:
 - o what happens if we perform a radiomic analysis per tract and not per parcel, i.e. considering only a single parcel per tract?
 - o Does the number of parcels vary from one subject to another? If so, how is this aspect managed in the radiomic analysis?
- The results are written rather elliptically. What are the DICE values in a nutshell?

- In classification tasks, 3-class problems are considered. Why then use the AUC, which is a performance metric mainly used in binary classification tasks? What are the results like if we switch to more conventional metrics such as accuracy, precision, recall, F1 score, etc?
- At no point (unless I'm mistaken) is the so-called classical tractometry used as a point of comparison for all the experiments explained. Why was no radiomic analysis ever carried out on the parcellation method used as a reference to see which had the greatest impact: radiomic analysis by parcels or the parcellation method? The term RadTract can be quite misleading at some points, particularly when it refers to parcellation methods comparison.
- Figure 4 is totally illegible
- I don't really understand the analysis by subset of features, which makes the whole paper more complex and adds a risk of random results to the story. Personally, I would have preferred a single analysis on a set of simple features (radiomic features from unfiltered images) to work in a smaller space, given the small number of subjects in the 4 cohorts.
- The regression task carried out on the clinical and demographic parameters lacks clarity: how was the cohort divided? why was no cross-validation carried out here? how was the data stratified between the training and test cohorts? How were the hyperparameters set?
- The description of figure 6 in the main body of the text lacks detail. Are the results obtained for the CC tracts representative of the conclusions for all the tracts?
- The discussion lacks content and critical insights. What about previous radiomic experiments in the literature on diffusion maps?
- Extraction of radiomic features: the input images for the ADNI cohort, for example, have different spatial resolutions. How is this point managed by the authors, as I don't see any methods for harmonising features being implemented?
- The spatial sampling used in the analysis is not mentioned at any point
- It's not clear to me how the ground truths of the classification task for tract parcellation were defined. This part needs to be made more explicit.

We want to thank the reviewers for their constructive, critical but also positive comments and suggestions. We rigorously revised our manuscript accordingly and conducted a large number of new experiments, and we are confident that we could address all points raised by the reviewers, and that we could improve the manuscript significantly. The most important changes introduced with this revision are the following:

(1) All reviewers pointed out that the manuscript contains two contributions, namely the radiomics approach and the new parcellation of the tract, and that these two contributions are not clearly separated. We agree that this is necessary to obtain a comprehensive and detailed picture of the new approach. Therefore, and following the suggestion by reviewer #2, we decided to focus purely on the radiomics aspect of RadTract in this work and to publish the new parcellation approach in a separate manuscript. Successively, all RadTract feature-based experiments were repeated using a centerline-based parcellation of the tract instead of the newly proposed approach. Besides disentangling the two contributions and simplifying the manuscript, this gives us more room to analyze and discuss the radiomics aspect of our work.

(2) We performed extensive new experiments, including both benchmark tractometry methods, and experiments to analyze the impact of different hyperparameters of the presented approach, namely the number of parcels as well as the number of automatically selected features. This hyperparameter analysis was performed on one of the four datasets (SCHZ). The optimal parameters were then used in the analysis of the other three datasets.

(3) Following the suggestion of reviewer #3 and to move away from the “proof of concept”-style of the first version of the manuscript, we dropped the analysis of individual feature subsets. Instead, we performed a single analysis on a set of simple features (radiomic features from unfiltered images).

(4) Following the suggestion of reviewers #1 and #2, we extended all experiments to include not only the Fractional Anisotropy (FA), but also Apparent Diffusion Coefficient (ADC), Radial Diffusivity (RD) and Axial Diffusivity (AD) maps.

Overall, the reduced feature set, the change of the parcellation method and the inclusion of additional parameter maps lead to results where the improvement introduced by RadTract are less pronounced than before, but they are still markedly improved over classic tractometry. In the following, we addressed all points raised by the reviewers individually.

Reviewer #1 (Remarks to the Author):

This paper introduces two separate innovations to the analysis of brain white matter tracts from in vivo diffusion MRI data. The first is in how white matter tracts are sampled for the purpose of

analysis. The other is in the use of a set of radiomic features for the purpose of machine learning analysis. Among its strengths is the fact that the paper includes analysis of several datasets on a range of different topics, showing potentially impressive improvements in performance. In addition, the paper includes an open-source software implementation of its main components. This should make the work more reproducible and extensible in the hands of others.

However, the paper also has some weaknesses. My main concern is that the results presented do not attempt to distinguish between the contributions of these two innovations. That is, it is not clear what are the relative contributions to improvements in performance from the use of radiomics features, and those that are due to the new parcellation scheme that is proposed. For example, one wonders how these algorithms would perform if the new parcellation scheme was used to produce FA profiles of the kind that is presented in Figure 1c and these profiles were used as features in the ML tasks presented in the Results section. The supplementary results suggest that at least in some cases, the parcellation by itself produces substantial improvements, and that the use of radiomics can help with additional improvement, but is by no means a dominant part of the improvement. Despite the inclusion of these in the supplement, I would like to see this question elevated with a more systematic analysis in the main results.

We agree that disentangling these contributions is important. Following the suggestion of reviewer 2, we decided not to include the new parcellation approach described in our original submission in this work, but rather publish this separately. Instead, we focus solely on the radiomics component of RadTract, i.e., we now compare classic tractometry (methods 1 and 2) to the new radiomics-based approach without the new parcellation but a centerline-based parcellation instead. This leaves more room to introduce and discuss the radiomics aspect of our work, and also simplifies the article.

Figure 1 and the Introduction section introduce important issues with two different existing methods for white matter tract parcellation (referred to as Method 1 and Method 2 in lines 74-76). However, the empirical comparisons in the results focus only on one of these (Method 2), leaving me unconvinced that the results are as damning as suggested to previous methods. To establish the novelty of this approach, it would be important to show that the differences are similarly large with method 1, which should more closely resemble the parcellation approach introduced here (e.g., in terms of Dice coefficients around the fanning parts of the bundles).

We agree that a comparison to static resampling tractometry (method 1) is required to obtain a comprehensive picture. We performed additional experiments on all datasets using this approach and extended the manuscript accordingly.

I also have some concerns regarding some of the statistical methods that were used:

1. Given the dimensions of the feature-set that is generated for every individual in every tract (e.g., 17 times 1,106 in the example shown in Figure 3) there must be a pretty large degree of

redundancy. The authors mention that they performed univariate feature selection down to 500 using scikit learn methods, but it is not clear how this was done. What does this feature selection do? Does it use the target categories for selection? Which of the thousands of features entered into the model are retained? Is it always a particular set of radiomic features? Specific locations in each tract?

This type of feature selection works by selecting the best features based on univariate statistical tests. The approach is implemented in Scikit-learn and documented here: https://scikit-learn.org/stable/modules/generated/sklearn.feature_selection.SelectKBest.html

In case of a classification experiments, this method computes the ANOVA F-value between each feature and the target variable. In case of a regression task, this is done by calculating the linear correlation of each feature with the target variable. Both approaches return the F-value and p-value for each feature. The F-value measures the difference in means between the classes or the linear correlation coefficient, for classification and regression respectively, while the p-value measures the significance of the difference/correlation. The higher the F-value and the lower the p-value, the more significant the feature is in predicting the target variable.

This approach for feature selection is supervised, meaning it does indeed use the target variables. Therefore, the feature selection is performed on the training samples only, not on the test samples.

The process is deterministic, meaning that given the same samples, the same features will be selected. Since we performed leave one out experiments, the selected features might be different across folds.

We extended the manuscript to describe this aspect of our work in more detail, and also extended the analysis of which features (feature type, location, parameter map) were selected and also deemed important by the random forest classifier for which task.

The code used for our classification and regression experiments, including feature selection, was further added to the RadTract python module and is publicly available.

2. The authors use a non-parametric test for comparisons of ROC curve results over ten rounds of LOOCV. To the best of my knowledge this approach violates the assumptions of the test. This is due to the fact that the 10 AUROC numbers obtained in these 10 experiments are not independent samples, because there is a high degree of overlap between each of the samples. The authors could instead use Delong's method for statistical comparisons of ROC curves. Fortunately, there is an open-source implementation of this algorithm here:

https://github.com/yandexdataschool/roc_comparison.

We agree that it is not sensible to use such a statistical test for this comparison. Delong's method is indeed a valid approach for comparing ROC curves. Nevertheless, it is designed to

compare only two ROC curves. Since we repeated all experiments ten times and also perform multi-class experiments, it is not straight forward to decide what to actually compare and comparing everything will not return any significant results. Furthermore, we are not investigating a clear medical or biological hypothesis but are instead focusing on illustrating that the new RadTract features have the potential to be more informative than the classic tractometry features. To this end, we analyze trends of the scalar AUROC values instead of the exact shapes of the individual ROC curves. After consulting with a statistician, we therefore decided that reporting statistical significance is not particularly sensible and helpful in this context.

3. A similar problem occurs for the Pearson correlation comparisons, where each of the 12,470 regression experiments is considered an independent experiment in the Wilcoxon test used, but these are highly dependent, because they are derived from repeated resampling of the same original sample.

We agree and as in the classification experiments, we decided that it is not sensible to perform statistical tests to highlight differences between the analyzed methods here. We updated the manuscript accordingly.

4. Another fact that bears explaining is how individual tracts seem to be able to predict better than the entire dataset. For example, it seems that parts of the CC make better classification prediction for the PPMI dataset than the entire dataset does.

This is a misunderstanding. The figure indicating this simply shows all conducted experiments per dataset in one plot, but it is not an experiment where one model was trained on features from all tracts. The figure was mere intended to make the detailed information contained in the per-tract plots more palpable and boil everything down to a single score per dataset. We clarified this aspect in the revised manuscript.

Finally, some comments about the framing of the paper, which could be addressed through revisions to the Introduction/Discussion: The paper introduces ideas from the nascent field of radiomics into tractometry. This is a very clever translation of recently-developed methods into a brand new application. However, I think that it would be worth explaining a bit more about (1) what radiomics is; I think that it is not a concept that has pervaded the MRI literature sufficiently yet to justify the implicit assumption that the target audience of this article has already heard about these methods (this reviewer, for one, had not heard about these methods before reading this manuscript), and (2) to explain what aspects of radiomics were used here.

We agree that more detailed information about the radiomics aspect of the work would be beneficial for our readers. Dropping the improved parcellation from this manuscript also gives us further room to focus on this aspect. The revised manuscript therefore features a more detailed introduction of the radiomics aspects relevant for this work.

In addition, I believe that the paper fails to address recent tractometry findings that do allow comparisons of individual participants and makes a different use of the information in the bundles. These findings were introduced in a paper by Chamberland et al. in Nature Computational Science volume 1, pages 598–606, in a method called "Detect". That method -- which does rely on "centerline" methods -- seems better suited to address the issue raised in lines 64-66 of the Introduction. A comparison of the new methods to Detect (at least at the conceptual level) would be important to establish some of the claims made in this paper.

We agree that Detect is definitely an approach that deserves to be discussed in the context of our work. Detect is an approach for unsupervised outlier detection based on centerline-tractometry, and can indeed be used for similar tasks as presented in our work. In fact, this work presents classification results on the SCHZ dataset also used in our work, where they report AUROC values of 0.64 ± 0.06 with their unsupervised approach (Detect) as well as supervised SV-classification results with AUROC values of 0.65 ± 0.13 , while we obtain results of 0.77 ± 0.01 (SCHZ tract ST_FO_left). Of course, the experimental setups are not completely comparable. What makes Detect an influential work, though, is not the overall classification performance, but the unsupervised-aspect of the approach. Furthermore, Detect is not a method that is competing with RadTract, but a method that could very well work with RadTract features instead of the currently used centerline-based tractometry features and might even yield improved results with this. So the basic idea of Detect rather corresponds to the Random Forest component of our work, but not to Radiomics Tractometry as a new class of features.

We extended the revised manuscript by a description and positioning of Detect in the context of our work.

I think that another limitation that the authors need to acknowledge is that they limited their analysis to fractional anisotropy and there may be important information in other image statistics. We extended all experiments to include not only FA but also features obtained on ADC, AD and RD maps. The results section was extended accordingly, including a paragraph describing which parameter map is most important for classification, which is typically indeed the FA.

Minor comments:

- Many of the Figures include font sizes that render the annotations completely illegible. All figures were refactored in the course of the revision and this issue was accounted for.

- Error bars used in Figures (e.g., in Figure 4b) should be explained. Are those standard boxplots?

Yes, we use standard box plots with 1.5 IQR whiskers. We clarified this in the revised manuscript.

- How were results chosen to be presented in Figure 4c? The choice of bundles across datasets seems inconsistent.

We did not present a subset of results, but the experiments were performed on these tracts specifically. The decision which WM tracts to analyze for which dataset was hypothesis- and evidence-based, depending on the respective pathology of the psychiatric and neurological disorder.

The CC serves as one of the most prominent tracts in the human brain that is responsible for the mediation of interhemispheric transfer, in terms of increased inhibition or reduced facilitation. Intact transcallosal functioning is essential for sustained attention, motor control, and synchronization of bilateral movements. For these reasons, CC plays a crucial role in the pathophysiology of all psychiatric disorders. In recent years, numerous systematic reviews have been published on the topic of structural and functional changes of CC in psychiatric disorders. A quick PubMed search using the terms "corpus callosum" AND "psychiatric disorders" with the specification article type "systematic reviews" on August 9th 2023 yielded a total of 54 references. Because of its crucial role in interhemispheric communication and transdiagnostic relevance, we decided to include CC-related tracts in our experiments on all four datasets. Further, besides CC, we also focused on the pathophysiologically most plausible WM tracts connecting crucial regions of the neuropsychiatric disorders represented in the four datasets.

We clarified this in the dataset descriptions of the methods section and described the rationale behind the choice of tracts in more detail for each dataset.

Reviewer #2 (Remarks to the Author):

Let me start by saying that bringing radiomics features in the world of tractometry is a great idea. Results on ADNI, CAT, PPMI, SCHZ seem promising and an improvement over current state-of-the-art tractometry. I do agree with authors that RadTract opens new potential for white matter analytics, especially given that this is also packaged as an open-source/open-science package.

Having said this, I have a few concerns with the current state of the manuscript that attenuate my enthusiasm. The paper is currently more a proof-of-principle than a final "product".

Major concern:

Diffusion MRI tractometry already suffers from the curse of dimensionality, i.e. N tracks, M metrics, R track-parcels, already leading to a massive set of $N \times M \times R$ features. Here, on top of that, radiomics features are added, 1106 radiomic features for each track, each metrics, each track-parcels. This is a monstrous number. At the moment, to account for this, authors use

- an automatic feature selection to only select 500,
- use the FA metric only in the experiments and results,

- use a subset of tracks of interest in the different databases

Justification for 500, FA-only, and subset of tracks per database are poor, especially on the different diseases (e.g. AD, PD, ...). You have selected mostly CC-related tracks. For me, authors have not figured out a RadTract "product" that would deal with ALL the data appropriately and have data-driven dimensionality reduction techniques implemented. RadTract is a nice prototype for future in-depth tractometry research.

We agree that dimensionality is an issue and that particularly the analysis per feature subset is more on the proof-of-concept as on the "product" side. In our revised work, we address all of those points, and we will give an overview of how we tackle the raised issues here:

(1) Justification for 500 features: this was indeed not justified but chosen rather arbitrarily. To give this a sound empirical basis, we used the SCHZ dataset to optimize these kinds of hyperparameters. Specifically for the used feature selection, we performed ten times repeated LOO classification experiments with $k \in \{50, 150, 200, 300, 400, 750, 1000, 1500, ALL\}$ and selected the best k for all other experiments. See Figure 6 in the revised manuscript.

(2) Justification for FA only: in our initial submission, we focused on the FA since it is the most extensively studied parameter map and therefore suitable to demonstrate that even such well-studied parameter maps contain more information than currently leveraged. Nevertheless, we agree that other parameter maps might very well hold additional information. Therefore, all experiments in the revised manuscript were performed with features from FA, AD, RD and ACD maps.

(3) Justification for the chosen subsets of tracts: We want to emphasize that our experiments were performed individually for each tract, not jointly on features obtained from all tracts. Therefore, for each experiment, the number of tracts N per experiment is always 1 and the total number of tracts included in our analysis has no impact on the number of features. We clarified this aspect in the revised manuscript.

That said, the actual choice of tracts was hypothesis- and evidence-based, depending on the respective pathology per dataset.

The CC serves as one of the most prominent tracts in the human brain that is responsible for the mediation of interhemispheric transfer, in terms of increased inhibition or reduced facilitation. Intact transcallosal functioning is essential for sustained attention, motor control, and synchronization of bilateral movements. For these reasons, CC plays a crucial role in the pathophysiology of all psychiatric disorders. In recent years, numerous systematic reviews have been published on the topic of structural and functional changes of CC in psychiatric disorders. A quick PubMed search using the terms "corpus callosum" AND "psychiatric disorders" with the specification article type "systematic reviews" on August 9th 2023 yielded a total of 54

references. Because of its crucial role in interhemispheric communication and transdiagnostic relevance, we decided to include CC-related tracts in our experiments on all four datasets. Further, besides CC, we also focused on the pathophysiologically most plausible WM tracts connecting crucial regions of the neuropsychiatric disorders represented in the four datasets. We clarified this in the dataset descriptions of the methods section and described the rationale behind the choice of tracts in more detail for each dataset.

(4) The number of features is monstrous: To make the whole setup more palpable and following the suggestion of reviewer #3, we limit our analysis in the revised manuscript to features obtained from the unfiltered images, which reduces the number of features from 1106 to 105 per metric and parcel. The resulting number of features is of course still large, but this is a typical issue in radiomics applications and well handleable using automatic feature selection approaches (see point 1).

While we dropped the features originating from the filtered images in the revised version of the manuscript for the sake of interpretability and simplicity, they might of course still hold valuable information. Such larger numbers of features are still handleable using automatic feature preselection, as demonstrated in our original submission, but investigating this will be one of the topics of future projects with RadTract.

All the described measures and new experiments were intended to drive the presented method more towards a “product” state. We are convinced that our experiments show nicely that RadTract is well usable in such a manner, that it will be a valuable tool for the whole research community and that our presented work provides a solid basis for future investigations using RadTract.

To further simplify future applications for classification and regression experiments with RadTract features, we extended the openly available RadTract code with this functionality, including the used feature selection.

Minor Concerns:

1) The new track parcellation approach is great. It seems to work. But for me, it deviates the high-level message of this work. Especially, in a Nature Comm publication. With the track parcellation included, the paper reads like a technical paper, with an incremental improvement in track sub-divisions. I would suggest publishing this part of the work in a methods paper or put less emphasize on it and send this part to supp. materials.

We agree completely and followed the suggestion to publish the new parcellation scheme in a separate manuscript. We repeated all experiments using a centerline-based parcellation as input for RadTract and revised the complete manuscript with a focus on the radiomics aspect, which should greatly improve the readability of the whole manuscript.

2) Because of 1) above, I don't find the paper particularly well-written or visually appealing.

3) As pointed above, if this paper is to be accepted, authors need to show how to include most dMRI metrics or DTI metrics in the pipeline, not just FA.

As described above, we followed this suggestion and extended our experiments and the manuscript accordingly.

4) Even with the new track parcellation, there are clearly some tracks that split in 2 or 3, where the track-parcellations, e.g. CC3, CC5 in the supp materials. Parts of the tracks, near the endpoints seem to clearly finish in different gyri. To me, this is a clear limitation of tractometry that hasn't be solved here. It is a multi-scale one. Should the track of interest be better defined? For example, defined at a scale where the endpoints cannot span more than one gyrus? This is an open question. But, somehow, I feel authors are ignoring it now. It should be added in a discussion.

This is true and definitely an issue. Nevertheless, this is a preprocessing step and depends strongly on the tool used to generate and segment the tracts. RadTract itself is agnostic towards the tool used to generate the tracts, and in this proof of concept study the most important aspect to ensure is that all methods use the same tracts as input. Since it is a popular approach that covers all tracts of interest for our work, we used TractSeg, but it would be completely fine to use any other approach. The impact of wrong or simply different tract segmentations by different methods on successive analysis methods is an interesting research question on its own and should undoubtedly be pursued. While it is probably relatively clear in the "finishing in wrong gyri" case you mentioned, the exact definition of many tracts is still the subject of intense discussions and different tract segmentation tools interpret them quite variably indeed. We included this aspect in the discussion of the revised manuscript.

Overall, great work. I'm being critical but I very much like the work.

Reviewer #3 (Remarks to the Author):

My experience is mainly in radiomic analysis of conventional MR images and my comments will therefore focus on this aspect. The aim of the proposed paper is to demonstrate, using 4 cohorts of moderate size, the added value of radiomic analysis performed on sub-segments of white matter bundle trajectories compared with conventional tractometry analysis. To this end, a new parcellation method based on an SVM technique is proposed, and compared with a conventional method based on an centroid-assignment strategy. From this new classification, sub-segments are extracted that serve as regions of interest for radiomic analysis. The analysis is original and the number of articles on the subject is limited. However, the analysis contains a number of grey areas that need to be clarified.

- The abstract is very macroscopic and does not at all highlight the originality of the proposed work or the methodology implemented.

In the course of the revision and the new focus of the work, we refactored the abstract completely, keeping in mind to convey the originality of the proposed work more clearly.

- In the introduction, only one of the assignment methods is properly detailed and illustrated (Figure 1), even though it is an important step in the paper. Assignment method number 1 deserves more detail. Moreover, it is not clear to me in the rest of the paper why only one assignment method (centroid-based) is used as a benchmark. Figure 1c is also completely incomprehensible as it stands.

We agree and extended all experiments to include both tractometry methods as benchmark for RadTract. In the revised manuscript, the old Figure 1 is no longer present.

- The number of parcels is automatically calculated per tract so as to obtain a number of voxels equal to 5 per parcel. This choice was made arbitrarily, but raises several questions:

o what happens if we perform a radiomic analysis per tract and not per parcel, i.e. considering only a single parcel per tract?

This is indeed an interesting question. We investigated this question on the SCHZ dataset, performing all experiments with multiple parcel numbers. Additionally, to the number of parcels used in our original submission, we performed experiments using twice the number of parcels, half the number of parcels as well as only a single parcel per tract. The best performance was indeed obtained using the originally proposed number of parcels based on a parcel thickness of 5 voxels, which was then used for all experiments on the other three datasets. The manuscript was extended to describe this hyperparameter optimization.

o Does the number of parcels vary from one subject to another? If so, how is this aspect managed in the radiomic analysis?

No, the number is constant across subjects and determined as the rounded average obtained from ten randomly chosen subjects. We clarified this in the manuscript.

- The results are written rather elliptically. What are the DICE values in a nutshell?

Following the suggestion of reviewer 2, we decided not to include the new parcellation approach described in our original submission in this work, but rather publish this separately. Instead, we focus solely on the radiomics component of RadTract, i.e. we now compare classic tractometry (methods 1 and 2) to the new radiomics-based approach without the new parcellation but a centerline-based parcellation instead. This leaves more room to introduce and discuss the radiomics aspect of our work, and also simplifies the article.

Consequently, the respective results paragraph about the parcellation and the corresponding dice scores was removed from the revised manuscript.

- In classification tasks, 3-class problems are considered. Why then use the AUC, which is a performance metric mainly used in binary classification tasks? What are the results like if we switch to more conventional metrics such as accuracy, precision, recall, F1 score, etc?

To calculate the multiclass AUROC, we used the One-vs-the-Rest (OvR) strategy, also known as one-vs-all, which consists in computing a ROC curve per each of the classes. In each step, a given class is regarded as the positive class and the remaining classes are regarded as the negative class as a bulk. The resulting AUROCs are averaged.

We opted for this performance metric, as it is well suited to globally compare approaches without a specific use-case in mind. The metrics mentioned by the reviewer are of course valid and perfectly adequate in certain settings, but they all require a threshold on the probabilities produced by the classifier. For this threshold and successively the resulting score to have any meaning, there has to be a certain reason how to choose it. For example, if a classifier decides if a patient should undergo surgery or not, the threshold has to be chosen based on the available data to realize a certain fraction of false positives or false negatives that might be ethically or medically acceptable. If such a rationale does not exist, choosing a threshold randomly will yield no valid performance indicator for the analyzed methods. This is the case in our work, where we want to know which method performs the best independent of the threshold. The AUROC provides just this information.

We explained our choice of metric in more detail in the revised manuscript.

- At no point (unless I'm mistaken) is the so-called classical tractometry used as a point of comparison for all the experiments explained. Why was no radiomic analysis ever carried out on the parcellation method used as a reference to see which had the greatest impact: radiomic analysis by parcels or the parcellation method? The term RadTract can be quite misleading at some points, particularly when it refers to parcellation methods comparison.

We agree, and this point is addressed in our revision by basing all RadTract experiments on the classic centerline-based parcellation and moving the new parcellation approach to a separate manuscript.

- Figure 4 is totally illegible

We refactored all figures keeping this in mind.

- I don't really understand the analysis by subset of features, which makes the whole paper more complex and adds a risk of random results to the story. Personally, I would have preferred a single analysis on a set of simple features (radiomic features from unfiltered images) to work in a smaller space, given the small number of subjects in the 4 cohorts.

We agree that focusing on a simpler feature set makes the entire concept more palpable and the results easier to interpret. We therefore followed the reviewer's suggestion and performed all experiments only based on the features calculated on the unfiltered images. This reduces the number of features per parcel and parameter map from 1106 to 105 and eliminates the need for

the analysis per feature subset. We further extended the feature importance analysis and report more detailed results about the importance of individual feature types, locations on the tracts and parameter maps for the individual datasets.

- The regression task carried out on the clinical and demographic parameters lacks clarity: how was the cohort divided? why was no cross-validation carried out here? how was the data stratified between the training and test cohorts? How were the hyperparameters set?

We agree that this is an unnecessary inconsistency. Therefore, we repeated the regression experiments in a leave one out way, as it was done for the classification experiments. The same hyperparameters of the random forest model as in the classification experiments were used for the regression experiments. Only the maximum tree depth of the RF models was manually set to 4 instead of the "None" default in order to avoid unnecessary long computation times. All other hyperparameters were left at default. No hyperparameter optimization was performed. We updated the revised manuscript accordingly.

- The description of figure 6 in the main body of the text lacks detail. Are the results obtained for the CC tracts representative of the conclusions for all the tracts?

The revised figure includes all analyzed tracts in the analysis.

- The discussion lacks content and critical insights. What about previous radiomic experiments in the literature on diffusion maps?

We extended the revised manuscript, specifically the introduction and discussion, to better position our work in the context of previous radiomics experiments on diffusion maps. Radiomics has shown to yield valuable insights and promising results for subject-level predictions in various tasks, such as automated diagnosis, patient stratification, risk assessment and response monitoring. In the context of brain imaging, radiomics has been used extensively for the analysis of tumors and also for studying psychiatric and neurodegenerative diseases. Nevertheless, the concept of radiomics has not yet found its way into the domain of tract-specific and along-tract WM analysis.

- Extraction of radiomic features: the input images for the ADNI cohort, for example, have different spatial resolutions. How is this point managed by the authors, as I don't see any methods for harmonising features being implemented?

All images were rigidly registered and resampled to the MNI-space FA template contained in the TractSeg package using MITK Diffusion. No further harmonization steps were performed. We clarified the latter in the revised manuscript.

- The spatial sampling used in the analysis is not mentioned at any point

We assume that the reviewer is referring to the image resolution. All images are resampled to an isotropic resolution of 1.25mm. If this is not the point referred to, we are happy to answer any further questions.

- It's not clear to me how the ground truths of the classification task for tract parcellation were defined. This part needs to be made more explicit.

Since we dropped the new tract parcellation approach from the manuscript with the goal to sharpen the manuscript's focus, this aspect is no longer relevant for the manuscript.

Nevertheless, this is an interesting aspect in general. As stated in the discussion of our original submission, there is no ground truth for tract parcellations. On a tract level, multiple expert-annotated datasets have been published recently, but no such annotations exist on a parcel level. An investigation of potential manual parcellation references might also include the question of more meaningful parcel borders based on biologically-founded subdivisions of WM tracts.

REVIEWER COMMENTS

Reviewer #1 (Remarks to the Author):

The authors have indeed made substantial improvements to the manuscript and the new version clarifies to a large degree the contributions of the methods that are proposed here. The additional explanation of radiomics and the untangling of the different contributions really does help understand what this paper contributes, and has improved the clarity of the paper and its contribution. However, with the improved clarity of the contributions I am afraid to say that I find that the contribution of radiomics to be rather less compelling than originally claimed. In light of the results presented here, I would still contend that the use of radiomics methods in analysis of diffusion MRI data is a clever and useful contribution, but I think that many of the claims of significance made in the article are overblown and need to be toned down quite substantially. I think that the "medical applications" in the title of the article should also be seriously reconsidered in light of the data.

For example, the authors contend that "Radiomics and tractometry are orthogonal approaches, in the sense that tractometry is focused on the localization of feature changes in a group study setting, while radiomics is focused on providing advanced biomarkers for predictive machine learning (ML) on a subject-level." (page 3). I have to say that I don't see evidence for this rather far-reaching conclusion in the data that is presented in the article (see more detailed comments below). In fact, in the revised article, it is hard to see any substantial advantage to the use of radiomic features relative to the baseline methods that are now used for comparison (and which indeed serve as a more appropriate baseline). The authors further claim that "we expect RadTract to spare critically ill and non-adherent psychiatric and neurological patients from lengthy diagnostic tests that are tedious, difficult to perform, and expensive" (page 4). This too is not substantiated by the data provided in the article, and I would suggest that the authors not make claims of this sort, unless they are clearly marked as extrapolative and highly speculative in nature.

Allow me to expand: the authors now provide more balanced comparisons between their novel method, radtract and methods that are the state of the art in the previous literature. The authors assert that radtract performs better than these methods in a truly impressive

number of really well-conducted experiments across multiple different datasets. However, the article now contains no formal comparisons that would tell the reader whether the methods differ more than would be expected by chance. In their response to previous comments, the authors mention that a statistician advised them not to perform comparisons between the results of the different methods. I have to disagree with this statistician's advice and ask that the authors provide quantitative assessments of the differences between methods. In particular, I would assert that without such comparisons, Figure 2 seems to demonstrate that for most tasks/tracts there is apparently no discernable difference between the three methods. One possible exception is in the ADNI AD classification, where there does seem to be a rather substantial advantage - it might be interesting to explore and understand why that is the case. Unfortunately, in all cases, areas under the ROC curve are also well below 0.8, with means in all cases below 0.7, a conventionally used threshold for tests to be considered better than "poor" (see e.g., Carter et al., 2016). This suggests that though these methods can reliably detect group differences, they would not be particularly useful for diagnostic purposes at the single subject level. As mentioned above, this is a major claim of the article, which should be substantially toned down in light of these results.

The results in Figure 3 also show that all of the methods used are roughly similar in terms of their overall performance, with correlations in most cases not reliably different from 0. These results are again too weak to be useful for any individual-level analysis, and claims about individual-level applicability of any of these methods needs to be toned down. A threshold of $r=0.15$ (2% variance explained) seems fine for demonstrating that there is a useful signal for group analysis, but is far from a threshold for applying these methods to individual participants. The improved performance in predicting individual age are rather heartening and demonstrate that radiomics methods do provide new kinds of information, but even so, a correlation of 0.6 (explained variance of less than 40%) is not remarkable in the literature on age effects on diffusion MRI measurements and an MSE of >70 years is also unremarkable relative to previous studies (I should note that the age range in the CAT dataset is not mentioned, and these numbers depend to some degree on the age range). An improvement to the presentation of the results in this section, which would make it more easily comparable to previous literature that uses machine learning in neuroimaging, would

be to use the coefficient of determination to assess model fit error, rather than the Pearson's correlation, as an assessment of variance explained. I don't expect that this will substantially improve any of the results to the point where they'd be considered strong diagnostic tests, but may help the interpretability of these numbers. For example, where there are substantial negative Pearson's correlations (e.g., for the GAF scale).

Another fly in the ointment of the claim that radiomic statistics provide novel useful information not available in previous methods comes from the results presented in Figure 4, where first order statistics are found to be the most informative features in all of the models. If I understand correctly, these first order statistics are also the same features that are used by the state of the art baseline methods, so this result by itself presents a challenge for a claim that radiomics methods uncovered some new and hitherto untapped information in diffusion MRI data. Again, the authors note that texture information was particularly useful in the ADNI dataset, and that made me wonder whether this fact is related to the substantial improvement observed in that dataset. If these facts were better understood and related to each other that may present an interesting criterion to determine in what cases radiomics approaches are more useful.

In summary, I still think that radiomics methods are a very useful addition to the arsenal of image processing tools available to researchers that use diffusion MRI data, but I think that the article currently makes some unsubstantiated claims about the utility of these methods and the magnitude of the advance over the previous state of the art. If that were to be corrected, I think that this contribution, together with the open-source software that the authors provide together with their article could gain significant traction in the research community and spur new kinds of investigations into brain tissue properties related to a variety of important and interesting phenomena.

Minor:

Caption of Figure 3 " $r > 1.5$ " should probably be " $r > 0.15$ " but see comment above about using coefficient of determination instead of Pearson's correlation coefficient. There is also one place in the text where " $r > 1.5$ " appears.

References:

Carter, J. V., Pan, J., Rai, S. N., & Galandiuk, S. (2016). ROC-ing along: Evaluation and interpretation of receiver operating characteristic curves. *Surgery*, 159(6), 1638–1645.
<https://doi.org/10.1016/j.surg.2015.12.029>

Reviewer #2 (Remarks to the Author):

I commend authors on their major revision. The revised manuscript truly addressed comments and issues raised by all 3 reviewers. Good job!

Reviewer #3 (Remarks to the Author):

I'd like to thank the authors for taking our comments into consideration, which has considerably improved the fluidity of the article. The contributions of these new analyses are now clearly highlighted, without any ambiguity about the conclusions.

My few residual comments are very minor:

- I think there is an error on the correlation coefficient mentioned in the body of the text and in the legend of figure 3 ($r > 1.5$).
- I would move Figure 6 and Table 1 to the results section, even if they are methodological results.
- I don't understand the conclusions of Table 1 for the 'static tractometry' case.
- In Figure 6, I would delete the cases $k > 400$ for centreline and static tractometries so that there are no ambiguities.

We want to thank the reviewers for again taking the time to provide constructive feedback and valuable comments. We addressed all points raised by the reviewers and revised the manuscript accordingly. For details, please refer to the point by point replies below.

Reviewer #1 (Remarks to the Author):

The authors have indeed made substantial improvements to the manuscript and the new version clarifies to a large degree the contributions of the methods that are proposed here. The additional explanation of radiomics and the untangling of the different contributions really does help understand what this paper contributes, and has improved the clarity of the paper and its contribution. However, with the improved clarity of the contributions I am afraid to say that I find that the contribution of radiomics to be rather less compelling than originally claimed. In light of the results presented here, I would still contend that the use of radiomics methods in analysis of diffusion MRI data is a clever and useful contribution, but I think that many of the claims of significance made in the article are overblown and need to be toned down quite substantially. I think that the "medical applications" in the title of the article should also be seriously reconsidered in light of the data.

For example, the authors contend that "Radiomics and tractometry are orthogonal approaches, in the sense that tractometry is focused on the localization of feature changes in a group study setting, while radiomics is focused on providing advanced biomarkers for predictive machine learning (ML) on a subject-level." (page 3). I have to say that I don't see evidence for this rather far-reaching conclusion in the data that is presented in the article (see more detailed comments below). In fact, in the revised article, it is hard to see any substantial advantage to the use of radiomic features relative to the baseline methods that are now used for comparison (and which indeed serve as a more appropriate baseline). The authors further claim that "we expect RadTract to spare critically ill and non-adherent psychiatric and neurological patients from lengthy diagnostic tests that are tedious, difficult to perform, and expensive" (page 4). This too is not substantiated by the data provided in the article, and I would suggest that the authors not make claims of this sort, unless they are clearly marked as extrapolative and highly speculative in nature.

While our results are very promising, and we can show that the proposed approach indeed significantly outperforms the state of the art (see below for discussion about statistical testing), we agree that some of the claims made in the prior versions should indeed be toned down in light of the new results. We adjusted the manuscript in several places according to this, and are in general describing RadTract as a promising research tool and as an important step towards subject-level prediction now, but we move away from claims such as medical applicability and direct usability for reliable subject-level predictions. Nevertheless, we believe that RadTract offers a very comprehensive and at the same time user-friendly solution not only for basic scientists but also for clinicians who want to conduct longitudinal studies on neuropsychiatric cohorts with neurodevelopmental and neurodegenerative mechanisms affecting white matter tracts.

Allow me to expand: the authors now provide more balanced comparisons between their novel method, radtract and methods that are the state of the art in the previous literature. The authors assert that radtract performs better than these methods in a truly impressive number of really well-conducted experiments across multiple different datasets. However, the article now contains no formal comparisons that would tell the reader whether the methods differ more than would be expected by chance. In their response to previous comments, the authors mention that a statistician advised them not to perform comparisons between the results of the different methods. I have to disagree with this statistician's advice and ask that the authors provide

quantitative assessments of the differences between methods. In particular, I would assert that without such comparisons, Figure 2 seems to demonstrate that for most tasks/tracts there is apparently no discernable difference between the three methods. One possible exception is in the ADNI AD classification, where there does seem to be a rather substantial advantage - it might be interesting to explore and understand why that is the case. Unfortunately, in all cases, areas under the ROC curve are also well below 0.8, with means in all cases below 0.7, a conventionally used threshold for tests to be considered better than "poor" (see e.g., Carter et al., 2016). This suggests that though these methods can reliably detect group differences, they would not be particularly useful for diagnostic purposes at the single subject level. As mentioned above, this is a major claim of the article, which should be substantially toned down in light of these results.

Following the reviewer's advice, we performed statistical tests for significance of our classification experiments using Delong's method for statistical comparisons of ROC curves. We show that the proposed approach ranks first among all compared methods with a statistically significant improvement in 14 tracts, while the two benchmark approaches achieve this in only 2 or even 0 tracts, respectively.

We handled the 10 repetitions for each case by testing each repetition independently, and a tract-level-result was deemed significant if the mean p of the repetitions was smaller than 0.05. In case of a multi class experiment, the tests were performed for each class independently using the One-vs-the-Rest (OvR) strategy and an improvement was deemed significant if $p < 0.05$ for at least one of the classes.

For increased transparency, we added figures showing the mean AUROC for the individual tracts and datasets, including 0.95 confidence intervals.

Further, while we agree that the mean AUROC values are not sufficient for reliable subject-level predictions, mean AUROCs obtained with RadTract are in fact better than poor (>0.7) in 5 tracts, as compared to 3 and 2 tracts for the benchmark methods.

The revised version of the manuscript has been extended to include the results of the statistical testing and the description of the performed tests. The major claims of the article were revisited in light of these additional results to make reassurance that they are backed up by our findings.

The results in Figure 3 also show that all of the methods used are roughly similar in terms of their overall performance, with correlations in most cases not reliably different from 0. These results are again too weak to be useful for any individual-level analysis, and claims about individual-level applicability of any of these methods needs to be toned down. A threshold of $r=0.15$ (2% variance explained) seems fine for demonstrating that there is a useful signal for group analysis, but is far from a threshold for applying these methods to individual participants. The improved performance in predicting individual age are rather heartening and demonstrate that radiomics methods do provide new kinds of information, but even so, a correlation of 0.6 (explained variance of less than 40%) is not remarkable in the literature on age effects on diffusion MRI measurements and an MSE of >70 years is also unremarkable relative to previous studies (I should note that the age range in the CAT dataset is not mentioned, and these numbers depend to some degree on the age range). An improvement to the presentation of the results in this section, which would make it more easily comparable to previous literature that uses machine learning in neuroimaging, would be to use the coefficient of determination to assess model fit error, rather than the Pearson's correlation, as an assessment of variance explained. I don't expect that this will substantially improve any of the results to the point where they'd be considered strong diagnostic tests, but may help the interpretability of these numbers. For example, where there are substantial negative Pearson's correlations (e.g., for the GAF scale).

As stated above, we agree that the results in general are not yet suitable for a subject level prediction. We toned down corresponding claims and mentioned this issue in the discussion.

The age range of the CAT dataset is 18 to 65, with a mean of 38.3 and a standard deviation of 11.4. We added this information for all parameters to Supplement S3.

While Pearson's correlation is more common in medical literature, we agree that the coefficient of determination (R^2) might be of interest for a certain target audience. As the reviewer pointed out, this indeed did not substantially change the results or any conclusions drawn from them. We added R^2 plots for all tracts and targets to supplement S3.

Another fly in the ointment of the claim that radiomic statistics provide novel useful information not available in previous methods comes from the results presented in Figure 4, where first order statistics are found to be the most informative features in all of the models. If I understand correctly, these first order statistics are also the same features that are used by the state of the art baseline methods, so this result by itself presents a challenge for a claim that radiomics methods uncovered some new and hitherto untapped information in diffusion MRI data. Again, the authors note that texture information was particularly useful in the ADNI dataset, and that made me wonder whether this fact is related to the substantial improvement observed in that dataset. If these facts were better understood and related to each other that may present an interesting criterion to determine in what cases radiomics approaches are more useful.

While it is the case that classic tractometry indeed relies on first order statistics, it uses only one feature (first order mean), while the proposed approach uses 18 first order features. Also, while the first order features individually have a higher importance, i.e., they reduce the impurity more than the individual texture features, the total reduction in impurity caused by first order features is much lower than the one of the first order features. This tells us that texture features are by no means less important than the first order features. We clarified this point in the revised manuscript. We also extended the manuscript to reflect on the particularly important role of texture features in the ADNI data experiments. This aspect, particularly in conjunction with the large improvement when using RadTract features, is in deed a very interesting phenomenon and will be investigated in future projects.

In summary, I still think that radiomics methods are a very useful addition to the arsenal of image processing tools available to researchers that use diffusion MRI data, but I think that the article currently makes some unsubstantiated claims about the utility of these methods and the magnitude of the advance over the previous state of the art. If that were to be corrected, I think that this contribution, together with the open-source software that the authors provide together with their article could gain significant traction in the research community and spur new kinds of investigations into brain tissue properties related to a variety of important and interesting phenomena.

Minor:

Caption of Figure 3 " $r > 1.5$ " should probably be " $r > 0.15$ " but see comment above about using coefficient of determination instead of Pearson's correlation coefficient. There is also one place in the text where " $r > 1.5$ " appears.

This has been corrected in the revised manuscript.

References:

Carter, J. V., Pan, J., Rai, S. N., & Galandiuk, S. (2016). ROC-ing along: Evaluation and interpretation of receiver operating characteristic curves. *Surgery*, 159(6), 1638–1645.
<https://doi.org/10.1016/j.surg.2015.12.029>

Reviewer #2 (Remarks to the Author):

I commend authors on their major revision. The revised manuscript truly addressed comments and issues raised by all 3 reviewers. Good job!

Thank you for your positive feedback! We are glad the revision was successful and appreciate your support.

Reviewer #3 (Remarks to the Author):

I'd like to thank the authors for taking our comments into consideration, which has considerably improved the fluidity of the article. The contributions of these new analyses are now clearly highlighted, without any ambiguity about the conclusions.

Thank you for your positive feedback! We are glad the revision was successful and appreciate your support.

My few residual comments are very minor:

- I think there is an error on the correlation coefficient mentioned in the body of the text and in the legend of figure 3 ($r > 1.5$).

This has been corrected in the revised manuscript.

- I would move Figure 6 and Table 1 to the results section, even if they are methodological results.

We agree and moved the description of the hyperparameter optimization to the results section.

- I don't understand the conclusions of Table 1 for the 'static tractometry' case.

We assume that the reviewer is referring to the fact that $k=100$ does not achieve the highest mean AUROC but is still chosen for the experiments. This is the case, because the mean AUROC might not be the highest in comparison to the other choices of k , but when taking the minimum (which we want maximized) and the variance (which we want minimized) into account, $k=100$ is the optimal choice (score to maximize = $\text{mean} * \text{min}/\text{var}$). We clarified this in the revised manuscript.

- In Figure 6, I would delete the cases $k > 400$ for centreline and static tractometries so that there are no ambiguities.

This has been corrected in the revised manuscript.